# Limited thermal tolerance in tropical insects and its genomic signature

Kim L. Holzmann[1✉], Thomas Schmitzer[1], Antonia Abels[1], Marko Čorkalo[1,2], Oliver Mitesser[3], Mareike Kortmann[3], Pedro Alonso-Alonso[1], Yenny Correa-Carmona[4,5], Andrea Pinos[6], Felipe Yon[7,8], Mabel Alvarado[9], Adrian Forsyth[10], Alejandro Lopera-Toro[10], Gunnar Brehm[4], Alexander Keller[6], Mark Otieno[1,11], Ingolf Steffan-Dewenter[1] & Marcell K. Peters[1,12]

Insects make up the majority of all animal species, with 70% occurring in the tropics[1], yet the impacts of warming on tropical insects remain highly uncertain[2]. This stems from sparse, taxonomically biased data on thermal tolerance of tropical insects and an incomplete understanding of the underlying physiological mechanisms[3]. Here we compared environmental temperatures with field-measured upper and lower thermal tolerance limits of around 2,300 insect species along Afrotropical and Neotropical elevational gradients and identified genomic signatures of thermal tolerance across the insect tree of life. We show that thermal tolerances do not proportionally track environmental temperatures but approach an asymptote in tropical lowlands. Insects at high elevations utilize plasticity to cope with rising temperatures, whereas lowland species have limited plastic abilities. Heat tolerance showed strong differences among insect orders and families, reflected in the thermal stability of proteins, suggesting that variation in thermal tolerance is founded in the fundamental protein architecture. Up to 52% of future surface temperatures and 38% of air temperatures in the Amazonian lowlands can cause heat mortality in half of the studied community. Our data suggest a limited capacity of insects in the Earth's most biodiverse regions to buffer future warming.

As global temperatures rise, insect diversity continues to decline at alarming rates in many regions[4]. More than 70% of all insect species are found in the tropics[1], where they are essential for ecosystem functioning[5]. As small ectotherms, insects are particularly vulnerable to increasing temperatures[6], yet our knowledge of their heat response is limited. Despite their immense diversity, data on thermal tolerances of tropical insects are sparse and heavily biased towards a few groups (for example, ants and fruit flies)[3]. Previous analyses suggest that upper thermal limits are relatively static along climate gradients with little plasticity[7–9], indicating that insects may have a low capacity to tolerate further warming[10].

Currently, tropical lowland places are experiencing increasing average and extreme temperatures due to anthropogenic warming[2], underscoring the urgency of understanding the ability of insect communities to respond to increasing temperatures and to reveal upper boundaries of heat tolerance. Besides heat, cold waves also challenge tropical animal communities and are expected to increase in intensity under future climate scenarios[11]. Species turnover, local adaptation to climate and evolutionary constraints influence thermal tolerance

limits in insects[12], but their relative importance across insect lineages remain unclear.

Heat tolerance is assumed to be strongly related to the ability of organisms to counter the destabilizing effects of high temperature on proteins[13]. Although production of heat shock proteins can counteract destabilization from heat to some degree, upper thermal limits of protein stability (measured as protein melting temperatures) are, in ectotherms, ultimate indicators of the threshold at which environmental temperature will have high costs and cause thermal injuries and mortality[10]. Although protein melting temperatures are commonly studied in structural biology, it is unclear whether they vary among insect orders and families, and how they are related to physiological heat tolerance.

Here, we experimentally quantified thermal tolerance ranges across 2,300 insect species spanning 242 families along Afrotropical (Kenya) and Neotropical (Peru) elevational gradients. We compared the relationship of critical thermal maxima ($CT_{max}$) to protein melting temperatures for several thousand proteins from more than 600 species across the insect tree of life. Finally, we predicted the thermal

[1]Department of Animal Ecology and Tropical Biology, Biocenter, University of Würzburg, Würzburg, Germany. [2]Department of Life Science, Tunghai University, Taichung, Taiwan. [3]Chair of Conservation Biology and Forest Ecology, Biocenter, University of Würzburg, Rauhenebrach, Germany. [4]Institut für Zoologie und Evolutionsbiologie mit Phyletischem Museum, Friedrich-Schiller University Jena, Jena, Germany. [5]Grupo de Entomología Universidad de Antioquia (GEUA), Universidad de Antioquia, Medellín, Colombia. [6]Faculty of Biology, Cellular and Organismic Networks, LMU Munich, Planegg-Martinsried, Germany. [7]Departamento de Ciencias Biológicas y Fisiológicas, Facultad de Ciencias e Ingeniería, Universidad Peruana Cayetano Heredia, Lima, Peru. [8]Instituto de Medicina Tropical, Universidad Peruana Cayetano Heredia, Lima, Peru. [9]Departamento de Entomología, Museo de Historia Natural, Universidad Nacional Mayor de San Marcos, Lima, Peru. [10]Andes Amazon Fund, Washington, DC, USA. [11]Department of Water and Agricultural Resource Management, University of Embu, Embu, Kenya. [12]Animal Ecology Group, BIOM, University of Bremen, Bremen, Germany. ✉e-mail: kim.holzmann@evobio.eu

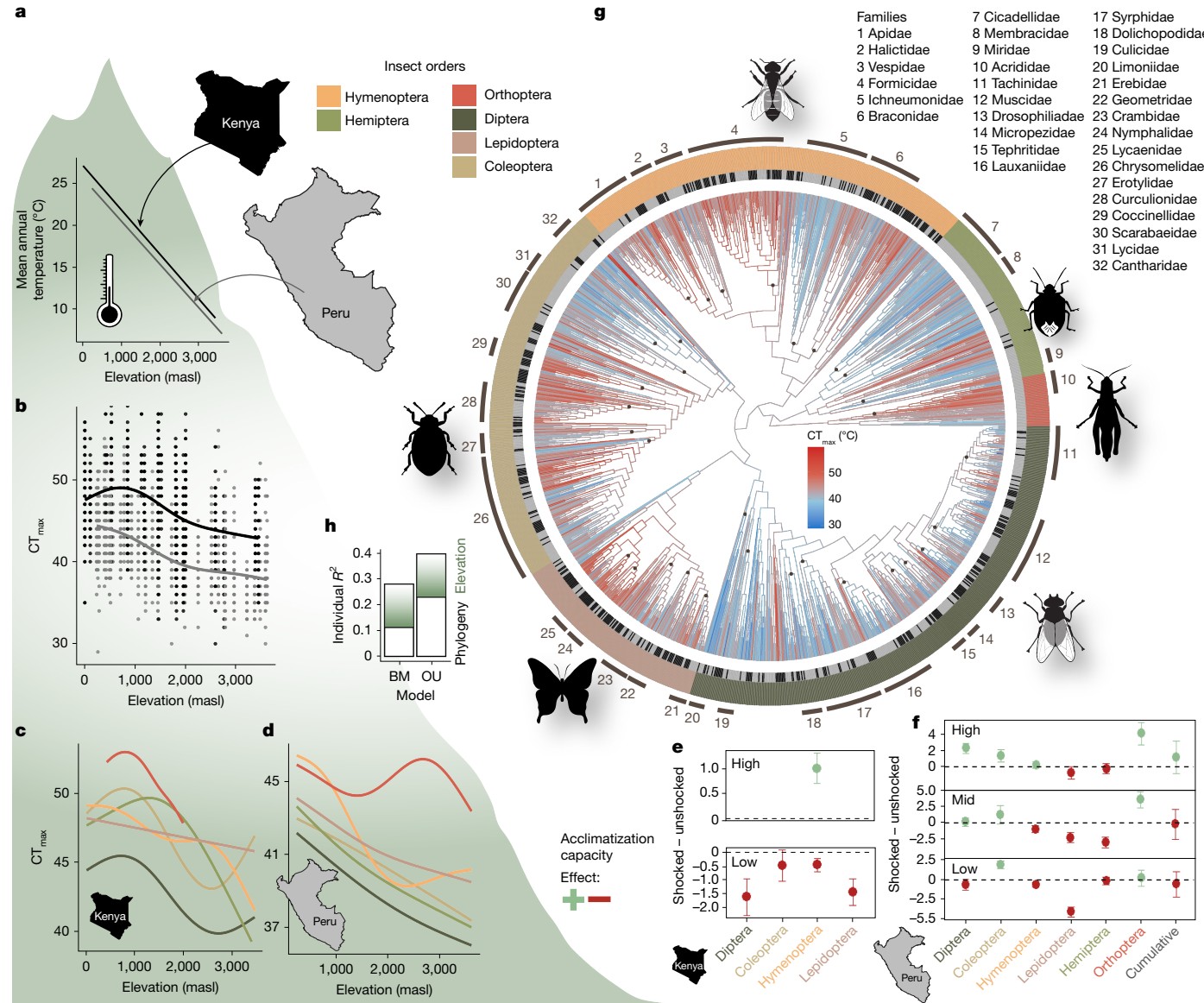

**Fig. 1 | Thermal tolerance and acclimatization potential across the insect tree of life. a**, Mean annual temperature decreased linearly and was slightly lower at the Neotropical (Peru) gradient compared with the Afrotropical gradient (Kenya). **b**, Field-measured critical thermal maxima ($CT_{max}$) of insects ($n = 3,229$) decreased along both gradients. **c,d**, This pattern was also visible along the Afrotropical gradient (**c**) and the Neotropical gradient (**d**) for each major insect order. Trend lines were calculated with generalized additive models with the smooth term parameter set to $k = 5$. **e,f**, For both the Afrotropical gradient (**e**) and the Neotropical gradient (**f**), a potential to tolerate higher temperatures after a heat shock treatment was evident at high elevations (high, >1,200 masl) but the effect decreased with elevation (mid, 600–1,200 masl) and became negative in lowland habitats (low, <600 masl) in most insect orders ($n = 777$ insects). Data are mean ± s.e.m. **g**, Phylogenetic supertree constructed by adding trees constructed from DNA sequences (from all sampled insects from which sequences could be derived) onto a family-level backbone tree with ancestral trait value reconstruction of $CT_{max}$. **h**, The variation in $CT_{max}$ was more strongly constrained by phylogeny than by effects of local temperature conditions (elevation), as indicated by variance partitioning. The Ornstein–Uhlenbeck (OU) model had better support than the Brownian motion (BM) model.

effects of current and future climates for Andean-Amazonian and East African insects.

## Heat tolerance and plasticity

We tested whether thermal tolerances of insect communities track temperature trends along elevation gradients (Fig. 1a) and whether there are indications of an upper limit in community responses to increasing temperatures. We measured the thermal tolerance of around 8,000 insects from approximately 2,300 DNA-barcoded species in the field using a common dynamic assay approach[14,15] (10 min acclimatization at 28 °C, ramping rate of 0.5 °C min[−1]). Thermal tolerance decreased from low to high elevation and was higher along the East African than along the Neotropical elevation gradient (Fig. 1b). This suggests stronger adaptations of climatic niches in tropical insects compared with species with a Holarctic distribution, whose upper thermal limits did not vary with elevation in a global study[16]. Long-term impacts of a less forested vegetation and higher exposure to heat in large parts of East Africa over the past million years have potentially shaped the greater heat tolerance of insect communities.

Thermal tolerance did not increase in direct proportion to environmental temperature along the elevational gradients. More precisely, for every 1 °C of increase in mean annual temperature, $CT_{max}$ changed approximately by 0.41 °C in the Neotropics and by 0.31 °C in

the Afrotropics. Trends in $CT_{max}$ and differences between geographic regions were evident in all major insect orders (Fig. 1c,d). Upper thermal safety margins, defined as the difference between environmental temperatures (here, mean annual temperature and the mean of the warmest month across a year) and thermal tolerance limits, increased significantly from the warm lowlands to the cold highlands (Extended Data Fig. 1). Along the East African elevation gradient, $CT_{max}$ increased linearly from high elevations to about 500 m above sea level (masl) and approached an asymptote at lower elevations (Fig. 1b). A tendency for a saturation curve was also observed in $CT_{max}$ along the Neotropical elevation gradient, but the pattern was less clear than along the Afrotropical gradient.

Insects may acclimatize to higher temperatures through stimulated changes in cell signals, metabolism and the production of heat shock proteins that stabilize proteins at stressful temperatures[17,18]. However, plasticity in upper thermal tolerance may be limited in insects that live in areas that are already exposed to very high environmental temperatures. To study plastic responses to environmental temperature, we conducted a heat shock experiment, exposing insects for 10 min to a sublethal temperature of 40 °C (for insects >2,700 masl: 35 °C) before exposing them gradually to increasing heat. The increase of $CT_{max}$ in response to a heat shock was on average 1.01 ± 0.31 °C (Afrotropics) and 1.10 ± 2.06 °C (Neotropics) at high elevations, matching results of a global meta-analysis testing plasticity in thermal limits of insects[8]. However, plasticity in $CT_{max}$ decreased from high to low elevations at both elevational gradients. In the warm lowlands, a sublethal heat shock did not increase thermal tolerance but led, on average, to reductions in $CT_{max}$ of insects (effect of heat shock on $CT_{max}$, −1.61 ± 1.04 °C in the Neotropics; −0.66 ± 1.59 °C in the Afrotropics; Fig. 1e,f). These results suggest that insect species at higher elevations have a modest potential to increase their heat tolerance by upregulating mechanisms that stabilize metabolic processes against heat. Our finding that a heat shock treatment negatively affects the thermal tolerance of insect species in the warm lowlands suggests that even without the experimental heat shock, protective metabolic mechanisms were already upregulated. Consequently, additional heat exposure from the heat shock treatment could have impaired the ability to withstand more heat[19].

We constructed a phylogenetic supertree by mapping DNA sequence-based trees from species to family level, derived from the insects tested for thermal tolerance, onto a family-level backbone tree of insect evolution[20] (Fig. 1g). Using $CT_{max}$ values of the surveyed species, we modelled ancestral trait values and assessed the phylogenetic signal in thermal limits across the insect tree of life. Visual inspection of the plotted $CT_{max}$ values on the phylogenetic tree and a formal phylogenetic correlogram (Extended Data Fig. 2) revealed a strong phylogenetic signal in $CT_{max}$ (Pagel's $\lambda$ = 0.76, $P < 0.001$; Blomberg's $K$ = 0.33, $P < 0.001$ and 10,000 randomizations), with positive correlations of $CT_{max}$ among related taxa[21]. Diptera showed, with few exceptions, generally low $CT_{max}$ values whereas high thermal tolerance measures characterized aculeate Hymenoptera (bees, ants and others) and Orthoptera. This suggests that thermal tolerance is a conserved trait which may have reciprocally influenced the evolution of microclimatic niches and optimization strategies within different insect orders: Whereas most Diptera are small organisms, which mostly depend on rapid development in temporary shaded and moist habitats[22], Orthoptera are often larger, have longer development times, and are often active in open, sun-exposed habitats[23].

We tested whether local adaptation and species turnover, reflected in elevational changes in $CT_{max}$, or phylogenetic constraints drive $CT_{max}$, and whether trait evolution follows a Brownian motion model, assuming random evolutionary drift, or an Ornstein–Uhlenbeck model, with stabilizing selection towards an optimum (Fig. 1h). In both models, elevation and phylogeny explained variation in $CT_{max}$ significantly, but the Ornstein–Uhlenbeck model was better supported by the data than the Brownian motion model (Extended Data Table 1). The long-term evolutionary optimum from the Ornstein–Uhlenbeck model of trait evolution corresponds to 42.20 °C ($\theta$ parameter). Variance partitioning of the preferred Ornstein–Uhlenbeck model showed that the phylogenetic relationship explained slightly more variation in $CT_{max}$ (partial $r^2$ = 0.23) than elevation (partial $r^2$ = 0.17; Fig. 1). Reconstructions of ancestral traits, a phylogenetic signal in thermal tolerance, the estimation of the long-term evolutionary optimum ($\theta$ parameter) and elevational trends in $CT_{max}$ were robust against sampling completeness, the exclusion of potentially paraphyletic families, problems in family-level assignments of individuals, the method for constructing family trees from DNA sequence information, and potential errors in branch length estimation (Extended Data Fig. 3 and Supplementary Table 1). Our data show, contrary to results of past meta-analyses[16], that insect communities show flexibility in responding to decreasing temperatures along elevation gradients. However, our results indicate upper limits in thermal tolerance set by evolutionary constraints, which could have already been reached by tropical lowland insects[9].

## Cold tolerance and plasticity

As insect responses to heat are costly, increases in $CT_{max}$ should be paralleled by reduced energetic investments in cold tolerance (critical thermal minima ($CT_{min}$)), such that thermal tolerance ranges ($CT_{max} - CT_{min}$) remain constant along temperature gradients[10]. To investigate thermal tolerance ranges, we additionally measured lower thermal limits along elevations in both geographic regions. The Neotropical data revealed only a slightly lower $CT_{min}$ of 5.53 ± 3.75 °C compared to the East African data with an average $CT_{min}$ of 6.04 ± 3.54 °C (Extended Data Fig. 4), so that the overall tolerance range was greater in the Afrotropics (40.25 °C versus 36.36 °C in the Neotropics). In both geographic regions, the thermal tolerance range across the whole insect community did not significantly change with elevation (Neotropics: estimate = 0.00036, $F_{1,23}$ = 1.785, $P = 0.195$; Afrotropics: estimate = 0.00003, $F_{1,13}$ = 0.004, $P = 0.951$). However, at the order level the thermal tolerance range significantly increased with elevation in the Neotropics in Hymenoptera (estimate = 0.00094, $F_{1,21}$ = 4.941, $P = 0.037$), Hemiptera (estimate = 0.00079, $F_{1,20}$ = 4.511, $P = 0.046$) and Orthoptera (estimate = 0.00251, $F_{1,14}$ = 10.39, $P = 0.006$), potentially indicating relaxed trade-offs between cold and heat tolerance in some lineages (Extended Data Fig. 5). We found no significant changes in thermal tolerance ranges in the Afrotropics. Along the Neotropical gradient, we tested the plastic capacity towards cold (cold hardening), by keeping insects for 10 min at 14 °C before gradually cooling them down. At high elevations, insects tolerated lower temperatures after the cold exposure, indicating the activation of cold hardening mechanisms[24], but this was not the case in the lowlands (Extended Data Fig. 4). The lack of a significant hardening potential in the lowlands might be due to the generally warm conditions making protection strategies for tolerating low temperatures unnecessary. However, a reduced cold tolerance could make insect communities in the lowlands sensitive to cold waves, which are predicted to become more frequent in the Amazon in the course of climate change[11].

## Thermal stability of proteins

Protein instability is assumed to be a major contributor to thermal coma and the eventual death of insects when exposed to heat[25]. Metabolic and physiological responses to heat are rarely studied but are assumed to be phylogenetically conserved[12]. Finding a strong phylogenetic signal in $CT_{max}$, we assessed whether variation in protein architecture among insect orders can explain variation in $CT_{max}$. We predicted the melting temperature ($T_m$) of a random selection of 1,000

proteins per species from published genomic data of 677 insect species (InsectBase 2.0; https://insect-genome.com) using the deep learning model DeepSTABp (https://rptu.de), which predicts protein thermal stability by estimating $T_m$ from protein sequence information[26]. We found—across all insect orders—the 25% most heat-sensitive proteins per species to have an average $T_m$ of 42.15 ± 0.73 °C (46.88 ± 0.61 °C for all proteins). The lowest heat-sensitive $T_m$ values were in Diptera with a mean $T_m$ = 41.17 ± 1.29 °C, and the highest $T_m$ values were in Orthoptera with $T_m$ = 43.40 ± 1.95 °C. Protein stability differed significantly across taxonomic orders ($F_{5,88637}$ = 1,951, $P$ < 0.001) and families ($F_{73,88569}$ = 226, $P$ < 0.001; Fig. 2) while controlling for species and protein identity in a linear mixed effect model. Predicted $T_m$ across insect orders and families were highly predictive of the observed $CT_{max}$ values (Fig. 2). This suggests that a part of the phylogenetic variation in organism-level $CT_{max}$ is due to fundamental differences in protein architecture across insect orders, which may have been optimized to different temperature levels in the early evolution of insects. Differences in $T_m$ among insect orders were consistently found for analyses including all proteins, the 25% proteins with the lowest $T_m$ per species and for analyses restricted to the 43 proteins shared in the randomly selected set of proteins across all orders (Extended Data Fig. 6).

## Thermal sensitivity under climate change

Although tropical insects already operate at very high temperatures, future warming of Amazonian or African ecosystems may push them beyond their physiological limit[6], as has been shown for other ectothermic taxa (for example, in amphibians[27]). We used $CT_{max}$ data to predict potential thermal injuries in insect communities caused by current (Fig. 3a,b) and future (Fig. 3c–h) surface (insects exposed to direct sunlight on surfaces) and air temperatures (insects in shaded above-ground environments). Notably, thermal tolerance limits are dependent on the interplay of temperature and exposure time[15]. Temperatures several °C below the measured $CT_{max}$ can already cause thermal injuries, resulting in heat coma if they accumulate over time[19,28]. Temperatures higher than the $CT_{max}$ determined by our protocol can lead to heat coma within time periods below 2 min. We calculated the knockdown time until coma ($t_{coma}$) for field-measured shaded air and surface temperatures[19]. Shaded air temperatures were continuously measured with loggers at our study sites (loggers were unavailable for the Afrotropical gradient, therefore the shaded air temperatures were modelled; Methods). Surface temperatures were derived from the Ecosystem Spaceborne Thermal Radiometer Experiment on Space Station (ECOSTRESS) sensor aboard the International Space Station, which provides unique high-resolution temporal and spatial measurements of land surface temperatures[29]. We calculated site-specific $t_{coma}$ values: first, for an average $CT_{max}$ per study site (using the median $CT_{max}$ of all measured insects per study site; Extended Data Fig. 7); second, for heat-sensitive insects (using the 25% quantile of $CT_{max}$ of all measured insects per study; Fig. 3); and third, for most heat-sensitive insects (using the 10% quantile of $CT_{max}$; Extended Data Fig. 7). Moreover, we calculated $t_{coma}$ for temperatures projected for three future climate change scenarios (SSP1-2.6, SSP3-7.0 and SSP5-8.5; Methods), adjusted for taking short-term variation in microclimatic temperature into account.

Under current climate, lowland insects in the Andean-Amazonian ecosystems are most affected by heat, where some measured surface temperatures can cause heat coma in less than 1 min (Fig. 3a,b). The threshold of heat coma in less than 1 min approximately matched our estimates of average protein melting temperatures of insects. Although air temperatures were relatively low, injury-causing surface temperatures were measured along the whole Afrotropical elevation gradient, probably owing to a high proportion of non-forested habitats. This suggests that during peak heat times, insects here need behavioural strategies to escape from heat[6]. Tropical butterflies, for

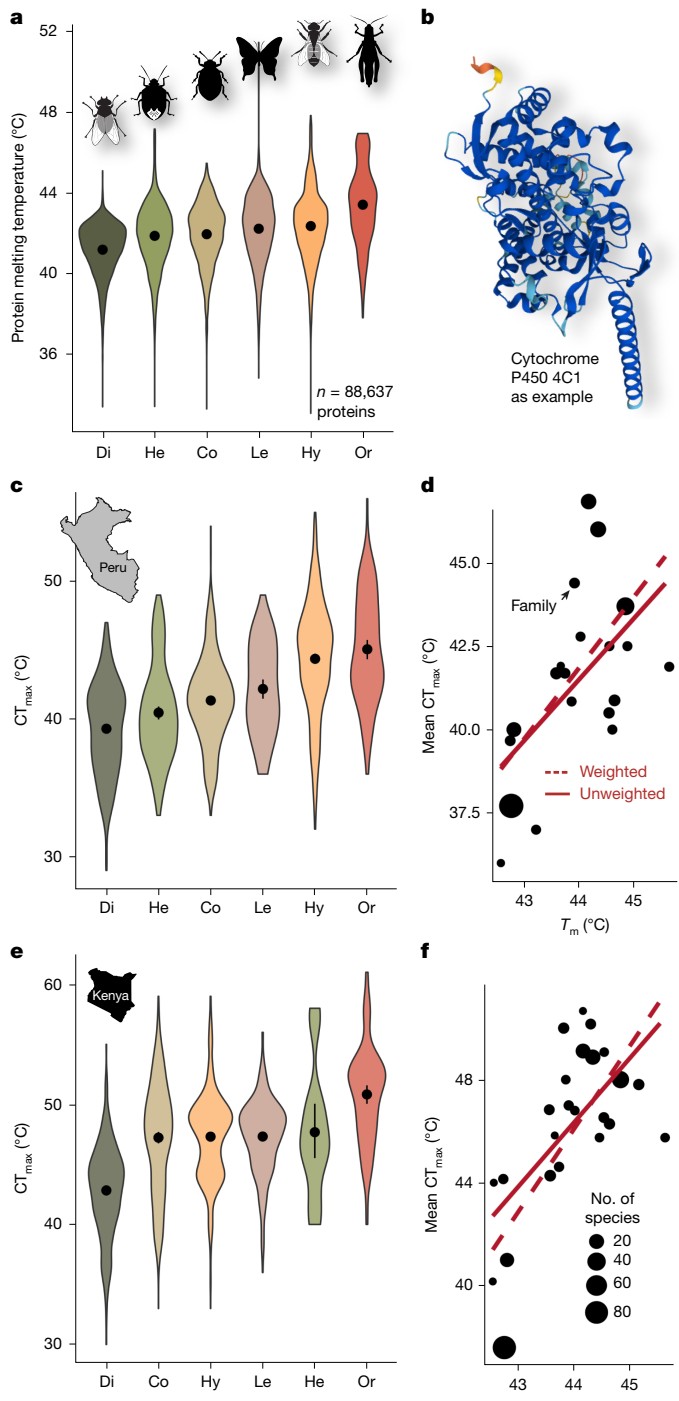

**Fig. 2 | Protein melting temperatures and critical thermal maxima.**
**a**, Predicted protein melting temperatures ($T_m$) across insect orders. Data for the 25% of proteins with the lowest $T_m$ per species are shown (a visualization of data of all proteins is shown in Extended Data Fig. 6). **b**, Melting temperatures depend on the stability of the structure of proteins, such as cytochrome P450 4C1 (example AlphaFold predicted structure). **c,e**, Experimentally tested critical thermal maxima ($CT_{max}$) of the six major insect orders ($n$ = 3,229 individuals) for the Neotropics (**c**) and Afrotropics (**e**). Dots and bars represent mean ± 95% CI and violin polygons show data density. **d,f**, Correlation of mean $CT_{max}$ and mean $T_m$ across all insect families from the Neotropics (**d**) and Afrotropics (**f**) for which both genomic data and data on $CT_{max}$ were available. The size of the dots is proportional to the number of species with genomic data per family. Trend lines show predictions of simple linear models and linear models weighted by the number of species with genomic data. Di, Diptera; Co, Coleoptera; He, Hemiptera; Le, Lepidoptera; Hy, Hymenoptera; Or, Orthoptera; ordered from lowest to highest mean.

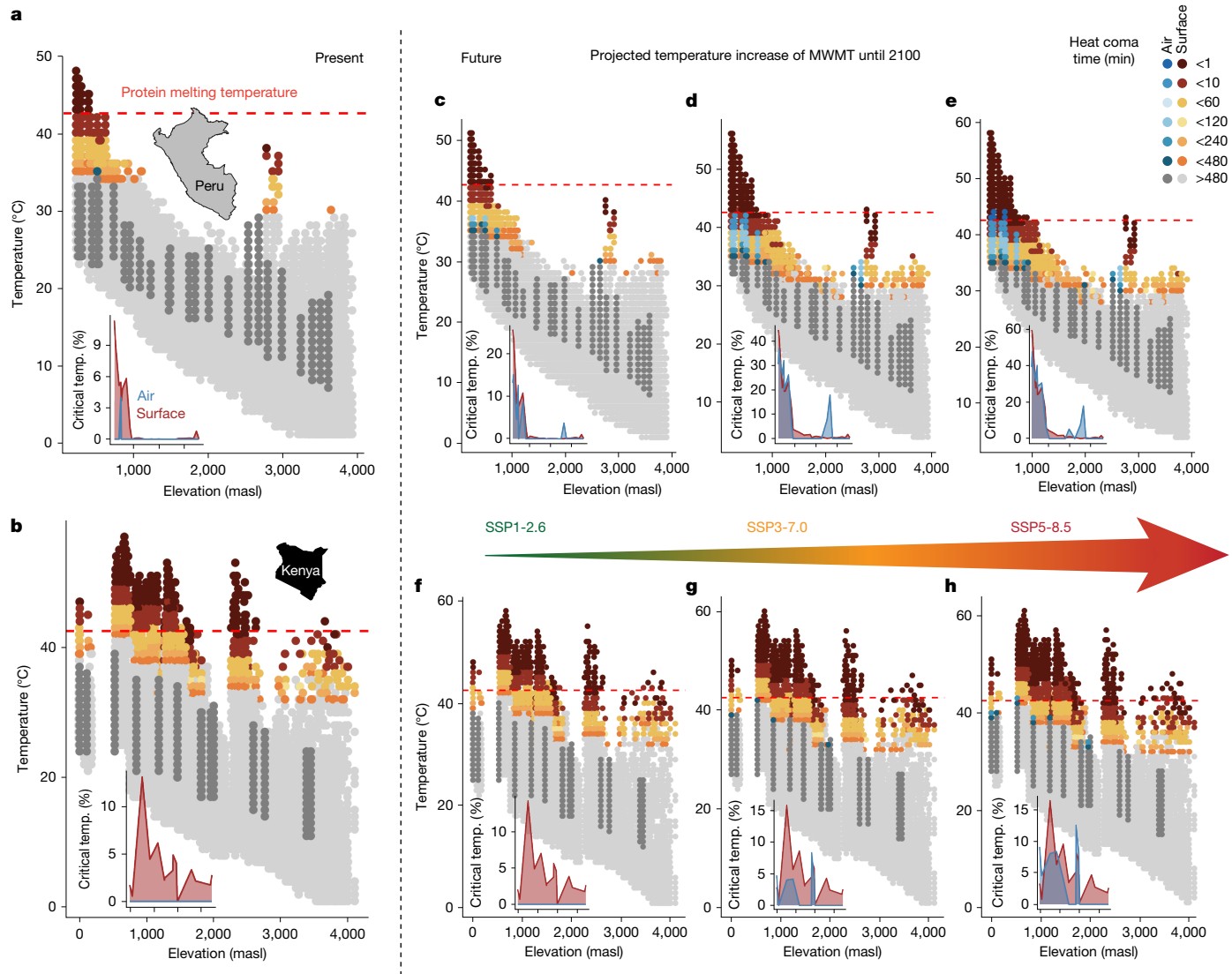

**Fig. 3 | Vulnerability of tropical insects to current and future temperatures.** Heat coma time for insect communities along Neotropical (**a**,**c**–**e**) and Afrotropical (**b**,**f**–**h**) elevation gradients, for present day surface and air temperatures (**a**,**b**) and surface and air temperatures predicted under three common climate change scenarios (climate model GFDL-ESM4 under SSP1-2.6 (**c**,**f**), SSP3-7.0 (**d**,**g**) and SSP5-8.5 (**e**,**h**)). In the main graphs, blue dots depict air temperatures that are high enough to cause thermal coma with an exposure of less than 8 h (critical temperature (temp.)), and dark grey dots indicate non-stressful air temperatures (shown are the highest 50% of all temperature measurements). Similarly, orange and red dots indicate critical surface temperatures, and light grey dots indicate temperatures below the threshold that causes substantial thermal injuries. The inset graphs show the percentage of all measured air (blue) and surface (red) temperatures that are critical for heat-sensitive insects (upper 25% of $CT_{max}$); ticks on the *x* axis represent the same elevations as those in the main graphs. The dashed red line shows the predicted mean protein melting temperature (42.15 °C), in line with the prediction that insects do not survive these temperatures.

instance, have shown a thermal buffering ability of up to 1 °C (ref. 30), and twig-nesting ants abscond their nests under heat stress[31]. We expect that avoidance of stressful temperatures by thermoregulatory behaviour is possible if: first, complex habitat structures exist that provide shade, which can buffer temperatures up to 4 °C (ref. 32); and second, shade air temperatures remain below the injury-causing threshold. In this respect, we found that almost all current air temperatures were below the temperatures that cause thermal injuries in relevant time periods. Furthermore, a high coverage by complex vegetation is evident in both elevation gradients (forest in the Neotropical gradient; forest, shrubland and savannah in the Afrotropical gradient) and provides a diversity of microclimatic habitats that can be used as heat shelter, reducing the risk of immediate overheating effects.

Climate change projections to 2100 (results reported below are based on the model GFDL-ESM4; for results based on a multi-model ensemble, see Supplementary Information) suggest that temperatures across the Neotropical gradient will surpass temperatures in the presently hotter Afrotropical gradient, with the most extreme temperature increases predicted for the Amazonian lowland (Extended Data Fig. 8). Taking upward elevational shifts of lowland species into account by conservatively assuming the $CT_{max}$ of the lowest plot to be valid across the whole elevation gradient, 20% of future surface temperatures predicted for the Neotropical lowlands (up to 300 masl) under SSP1-2.6 will be high enough to cause heat coma in half the insect community within 8 h (hereafter called 'critical temperature'). Under SSP3-7.0, the proportion of temperatures above the critical temperature increases to 39% and under SSP5-8.5 it is 52% (Fig. 3c–e). Notably, using a median

$CT_{max}$ of lowland insects, 9%, 29% and 38% of all future air temperatures in the Neotropical lowlands are predicted to be critical under SSP1-2.6, SSP3-7.0 and SSP5-8.5, respectively. For the more heat-sensitive insects (based on the 25% quantile of $CT_{max}$), 26%, 45% and 59% of all surface temperatures under SSP1-2.6, SSP3-7.0 and SSP5-8.5, respectively, are predicted to be critical (results for most heat-sensitive group in Extended Data Fig. 7). These values are conservative estimates, assuming that future surface temperatures warm at similar rates as air temperatures (Methods). In total, 15% (SSP1-2.6), 37% (SSP3-7.0) and 47% (SSP5-8.5) of all future air temperatures will be critical for the group of more heat-sensitive insects (Fig. 3c–e). These results suggest that under less optimistic climate change scenarios, heat refugia for Amazonian insects will shrink as future daytime temperatures reach critical levels. However, the response of insects to climate warming is highly complex and experimental assays can only provide a proxy to predict heat tolerance, as populations may decline at temperatures even lower than those that cause heat coma owing to sublethal heat injuries[19] and because extreme heat events can lead to increased failure of biological processes before the $CT_{max}$ of an organism is reached[33].

In East Africa, under all three climate change scenarios, the predicted increase in temperature is considerably less than the values predicted for the Andean-Amazonian elevation gradient, leading to slight increases in the proportion of surface temperatures that can be regarded as critical. Air temperatures in East Africa are not critical under SSP1-2.6, whereas under SSP5-8.5, up to 13% of the air temperatures are high enough to cause heat coma in more heat-sensitive insects (Fig. 3f–h and Extended Data Fig. 7).

## Conclusion

Our findings—(1) a saturation in upper thermal tolerance from cold highlands to warm lowlands; (2) narrow thermal safety margins towards lower elevations; (3) no evident plasticity or hardening potential in lowlands species; (4) support for phylogenetically constrained evolution of upper thermal tolerance limits; and (5) a high percentage of proteins with melting temperatures approximately in the range of current surface and future air temperatures—suggest that upper limits in the evolution of thermal tolerance traits in tropical lowland insects have nearly been reached[34,35]. Although these data cannot entirely exclude the possibility of evolution of protein structures to adapt to higher temperatures or the upregulation of mechanisms that stabilize protein structure under heat (such as expression of heat shock proteins), these adaptations would entail high costs and are unlikely to be feasible for the vast number of essential proteins required to sustain metabolism[10]. Under current climatic conditions, species along the Afrotropical elevation gradient appear closer to their physiological thermal limit, but species along the Neotropical elevation gradient are likely to be more vulnerable under climate change. In the Amazon, temperature anomalies are projected to be much higher and might escalate owing to feedback loops between drought and forest loss[36]. In the Amazonian lowlands, both surface and air temperatures are predicted to surpass current upper thermal limits of insects. Without strategies to escape from heat by shifting to higher elevations or other cool refugia[37] by diapause or by finding shelters from heat[38] many insects will experience thermal death. Both heat waves and cold waves will challenge insect communities in the next decades[11]. Intact tropical rainforests that provide shade and thermal habitat complexity will be essential to buffer the effects of further climate warming. However, opportunities to avoid heat stress are likely to diminish as tropical rainforests become increasingly open through deforestation, logging and tree mortality induced by climate change[36]. Forest connectivity to allow lowland insects species to migrate to higher elevations will be crucial for the survival of insects in the world's most biodiverse regions.

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

## Methods

### Study area

In Peru, the study was carried out along an elevational gradient from 245 masl to the tree line at 3,588 masl in the Andes of south-east Peru (Kosñipata valley), with continuous and mostly undisturbed wet rainforest/cloud forest. The climatic gradient has three seasonal periods with a wet season (November to March), a dry season (May to July) and austral spring (September and October)[39]. Mean annual temperatures range from 24.3 °C in the lowlands to 6.7 °C at 3,600 masl (Fig. 1). Mean annual precipitation levels are high with >1,500 mm per year along the whole gradient, peaking at around 1,500 masl with ~5,000 mm (ref. 40). Research was conducted on 26 study plots of approximately 100 m × 100 m in ~250 m elevation intervals and seven field stations distributed along the gradient. Four of the 26 plots were located inside Manú National Park. Nine of the plots matched the long-term research plots of the Andes Biodiversity and Ecosystem Research Group (ABERG) project[41].

In Kenya, the study was carried out along an elevational gradient from 11 masl at Watamu to 3,450 masl at Mount Kenya including forests, woodland, scrub and grassland in natural and semi-natural habitats. The climate is mostly semi-arid in the lowlands to humid at higher elevations[42] and characterized by seasonality in precipitation. Two rainy seasons occur from March to May and from October to December, a distinct season during the boreal summer from June to September, and a pronounced dry season in January and February[43,44]. The forested parts of the Taita Hills region and Mount Kenya from ~1,300–2,500 masl are characterized by a tropical montane forest climate with generally high humidity and constant high precipitation (>1,500 mm). Mean annual temperatures range from 26.2 °C at the lowest plot to 8.9 °C at the highest plot (Fig. 1). In total, 15 study plots were selected in similar elevation intervals along the elevation gradient.

### Insect collection

Insects of all major orders (mainly Coleoptera, Diptera, Hymenoptera and Lepidoptera; additionally, Hemiptera and Orthoptera) were collected in the Neotropics ($n$ = 4,690) in three seasons (September to December 2022, April to August 2023, and September to December 2023) with sweep nets in the understory of all study plots. In the Afrotropics, insects ($n$ = 3,164) were collected in one season (March to June 2023), applying the same method. We stored live insects in Eppendorf tubes with moist tissue and sugar solution, protected from the sun, and transported them back to the field station, where thermal tolerance measurements of all collected individuals were carried out on the same day. Insect collection was focused on taxonomic breadth covering six major orders rather than on individual species. Note, that our results represent the more common insect communities. In the Supplementary Methods, we provide extensive robustness analyses to assess the effects of incomplete sampling or modifications to the phylogeny.

### Measuring thermal limits

We measured critical thermal limits (CT) by exposing the insects in individual plastic tubes (2, 5 or 50 ml, depending on body size) to decreasing ($CT_{min}$) or increasing ($CT_{max}$) temperatures using a programmable thermoblock (Eppendorf Thermostat C)[7,45,46]. Each individual was only tested once. To avoid the effects of starvation, each tube was equipped with a piece of paper towel moistened with sugar water. For acclimatization to a common baseline, insects were first exposed to the starting temperature of 28 °C for 10 min. We then increased or decreased temperatures by 1 °C every 2 min—that is, a ramping rate of 0.5 °C min$^{-1}$—following standardized methods[14]. After each 2-min interval, we checked all insects for mobility. The temperature at the point of lost mobility, even after tapping or gently shaking the tube, was noted as the upper or lower thermal limit ($CT_{max}$ or $CT_{min}$)[14,15]. After the

test, insects were stored in 96% ethanol for later genetic barcoding and specimen documentation. Since we did not directly measure the body temperature of the insects, $CT_{min}$ and $CT_{max}$ values determined with the above-mentioned protocol may not perfectly reflect body temperatures. Nevertheless, owing to the small size and their high surface-area-to-volume ratio, insects equilibrate to environmental temperatures quickly, typically within seconds[6].

All thermal limit data were originally stored as csv or xlsx files (Microsoft Excel v.2502) and imported into R v.4.3[47]. Since four different observers measured CTs in the field, we first tested for potential observer bias by evaluating the $CT_{max}$ of lab-reared ants of one colony. Each observer measured $CT_{max}$ of ten individual workers of the colony under the same conditions. We found no significant difference in mean $CT_{max}$ across any of the observers (ANOVA, $F_{3,36}$ = 1.576, $P$ = 0.212).

For investigating CT patterns along elevation gradients, we applied generalized additive models with elevation as a smooth term explanatory variable. We restricted the smooth term parameter $k$ to 5 to avoid overfitting[48,49]. Thermal tolerance ranges were calculated by subtracting mean $CT_{min}$ from mean $CT_{max}$ for each study plot. We checked the trend of thermal ranges along elevation with an ordinary linear model. We additionally applied the same models for each major insect order. Thermal safety margins were calculated for both datasets by subtracting plot-specific variables of mean annual temperature and mean daily maximum air temperature of the warmest month from the $CT_{max}$ values. Both temperature data variables were derived from the CHELSA data base (bio1 and bio5 of the BIOCLIM+ dataset)[50].

As a test for plastic responses, a subset of insects was first exposed to heat (40 °C, $n$ = 777 insects) or cold (14 °C, only in the neotropics, $n$ = 490 insects) shock for 10 min prior testing, replacing the 28 °C acclimatization period. The sublethal temperatures for the heat shock were identified in pre-experiments. In the Neotropics up to an elevation of 2,700 masl, insects generally survived a 10 min shock of 40 °C; at higher elevations 35 °C was chosen as shock temperature, because a large proportion of the pilot test insects did not survive a 40 °C shock. After the shock, the standard protocol continued from 28 °C with temperature changes in 2-min intervals.

Plastic capacities were analysed by comparing CT measurements between individuals that were exposed or not exposed to a heat or cold shock before CT measurements. We added heat shock or cold shock (yes/no) as a variable for $CT_{max}$ or $CT_{min}$, respectively, and calculated the means for both groups. Data were additionally grouped by elevation level (lowlands: <600 masl, mid: ≥600 masl and <1,200 masl, high ≥1,200 masl), because we expected heat shock effects to vary depending on elevation. We hypothesized that a thermal shock would increase tolerance—that is, have a positive effect. Therefore, for $CT_{max}$, effect calculations were done by subtracting the mean thermal tolerance of the control group from the mean thermal tolerance of the group exposed to a heat shock ('shock yes' – 'shock no'). For $CT_{min}$, the calculation was done in reverse ('shock no' – 'shock yes'), so that the effect direction remains the same, with a positive effect indicating an increased tolerance level.

### Insect identification and phylogeny

All insects were morphologically sorted into orders and, where possible, families, and delimited into species-like units based on individual DNA barcoding. For this, we sampled tissue from all specimens that were tested for thermal tolerance and positioned them in a well filled with 30 µl of absolute ethanol (99.9%) in a 96-well microplate. Sequencing preparation and conduction was done at the Canadian Centre for DNA Barcoding (CCDB). Libraries were created for DNA barcoding (standard 658 bp COI (mitochondrial cytochrome c oxidase subunit I) barcoding) using SMRT sequencing technology on a PacBio Sequel IIe[51]. All genetic sequences were uploaded together with specimen photographs and sampling information to BOLD (database code DS-A2TP; https://www.boldsystems.org/). The COI sequences had an average length of 557 bp

with a minimum overlap of 303 bp (74% of all COI sequences were larger than 500 bp and 50% larger than 600 bp).

We applied the BOLD sequence cluster tool which uses the refined single linkage (RESL) algorithm with a pairwise distance model to sort all insects into unique operational taxonomic units (OTU), which we used as species-like units (called 'species' in the text)[52]. Contaminants, records with stop codons and sequences with a length below 300 base pairs were excluded. In total, 4,300 barcoded individuals were included in the final phylogenetic analyses of which 2,330 were unique OTUs, resulting in an average of 1.8 individuals per OTU. The $CT_{max}$ data included 2,246 measurements with 1.9 individuals per OTU, the $CT_{min}$ data included 1,849 total observations with a mean of 2 individuals per OTU. We used the BOLD ID engine (minimum of 80% similarity to databased sequences) to allocate all sampled insects to family level[53].

We used R to create a phylogeny based on a family-level backbone tree and DNA sequences[20]. For this, we downloaded an insect family-level backbone tree[54]. For each family, a separate subtree was calculated first, using the AlignSeqs function from DECIPHER[55] and a maximum likelihood model (TreeLine function). Statistical support for each branch is provided by aBayes values (see data repository). If a family consisted of only one OTU, a 'tree' with a single branch of fixed length was constructed. Next, all subtrees were added to the ultrametric backbone tree using the bind.tree function from the ape package[56]. Branch lengths were calibrated with the bladj function from phylocom[57], that estimates node ages based on fossil calibration points while unknown ages are evenly distributed between known nodes[54]. For clearer visualization (Fig. 1h), terminal tip heights were equalized with the forceEqualTipHeights function from ips[58]. Note that while this gives the appearance of an ultrametric tree, the branch lengths do not represent absolute time, but the result is a phylogeny with relative branch lengths. All scripts and details of phylogeny construction can be found in[20].

We reconstructed ancestral trait values of thermal tolerances using the fastAnc function from phytools[59], plotted them on the phylogenetic tree with ggtree[60], and calculated a phylogenetic correlogram using phylosignal to test for significance of the phylogenetic signal—that is, if trait values of related OTU are more similar (or dissimilar) than expected by chance[21]. The phylogenetic parameters Pagel's lambda and Blomberg's $K$ were tested with the phylosig function and 10,000 randomizations. To disentangle the effects of adaptation or acclimatization to a local climate from those of phylogenetic relatedness, we applied a phylogenetic regression using the phylolm package, and function of the same name, with elevation as predictor and phylogenetic relationships entering a covariance matrix of the model either based on the assumption of a Brownian motion and an Ornstein–Uhlenbeck model of trait evolution[61]. We compare the Ornstein–Uhlenbeck and Brownian motion models since they represent two mechanistically contrasting hypotheses about trait evolution and are most commonly used in literature[9]. Under a Brownian motion model, traits are assumed to have evolved under random evolutionary drift, under an Ornstein–Uhlenbeck model with stabilizing selection towards an optimum. We calculated the evolutionary optimum for $CT_{max}$ under an Ornstein–Uhlenbeck model—that is, the parameter $\theta$, using the fitContinuous function of the geiger package[62]. We compared model performance using the Akaike information criterion[63]. Variance partitioning to extract partial $r^2$ values was conducted on these models using the phylolm.hp package[64].

In order to verify the dependence of results concerning elevational trends in CT, on the reconstruction of ancestral traits, the estimation of a phylogenetic signal and of an upper boundary of $CT_{max}$ (Ornstein–Uhlenbeck model), we conducted an extensive set of robustness analyses considering various modifications of the phylogenetic tree and various data subsets, which is described in the Supplementary Information, with results presented in Extended Data Fig. 3 and Supplementary Table 1.

## Protein stability prediction

We predicted the thermal stability of proteins applying the deep learning model DeepSTABp[26]. It uses a transformer-based protein language model to extract sequence embeddings, which are analysed with advanced deep learning techniques for large-scale protein stability predictions[26]. To cover proteins from species across all insect orders, we downloaded all available genomes (protein format) from 677 insect species from InsectBase 2.0[65]. This data covered 20 orders, 158 families and 457 genera. Next, in R, we imported all genomic data translated into amino acid FASTA files and then randomly selected 1,000 proteins per species, a trade-off to receive as many overlapping proteins between species as possible while keeping the following analyses at a feasible computation time. We set up a local version of DeepSTABp, to predict melting points for this large number of proteins. Anaconda PowerShell was used to create a conda environment and Python programming language (v.3.13) to run the script[66]. In DeepSTABp, we set growth conditions to 'cell' and a default temperature of 22 °C (ref. 26). For testing robustness of results, we conducted analyses with data from all genomes, only high-quality genomes (BUSCO > 89.9, N50 > 300 kb), and only for proteins which were covered by all insect species. Furthermore, for comparison we used data of all proteins and of the 25% proteins per species that showed the highest thermal sensitivity ($n$ = 88,643 proteins from 677 species).

To investigate the underlying, structural mechanism of heat tolerance in insects, we predicted $CT_{max}$ by protein melting temperatures $T_m$. Differences in $T_m$ across orders and families were tested with mixed effect models (lme), setting species-specific protein identity as random term. For the comparison of $T_m$ with field-measured $CT_{max}$, we calculated mean $CT_{max}$ values for each family of the Afrotropical and Neotropical data. We calculated the 25% quantile of $T_m$ for each species (that is, a proxy for an average $T_m$ of temperature sensitive proteins) of the genomic dataset and then averaged these values among all species per family. Using a linear model (lm), we tested the relationship between $T_m$ of the temperature sensitive proteins (average of the 25% quantile of $T_m$) and the mean $CT_{max}$ of families. Here, we first calculated an unweighted linear model and, second, a model weighted for the number of species with genomic data in each family.

## Climate data

On all Neotropical plots, one TMS-4 soil and one air temperature logger (TOMST) recorded air temperature at 15 cm and 2 cm above the ground, as well as soil temperature and moisture at −6 cm depth in 15-min intervals for approximately one year (September 2022 to December 2023)[67]. Air temperature was additionally measured at 1.5 m height with iButton sensors (Analog Devices) in 240 min intervals. The sensors were protected from rain and sunlight using white plastic dishes (diameter 18 cm)[68]. Air temperatures at 1.5 m ($T_{air\,at\,1.5\,m}$) were highly predictive of the temperature at 15 cm height ($T_{air\,at\,15\,cm}$)—$R^2$ = 0.999, $T_{air\,at\,1.5\,m}$ = 0.26673 + 0.99845 × $T_{air\,at\,15\,cm}$—and due to the denser sampling intervals, data of the TMS-4 sensor were used for final statistical analyses. At one plot (ID 056), the TMS logger could not be recovered due to fallen trees; therefore, the mean annual temperature from iButton data was used to predict $T_{air\,at\,15\,cm}$ for this plot instead. For the East African study plots, we did not have temperature loggers available but retrieved comparable data by modelling shaded air temperatures using NicheMapR. We simulated hourly temperatures at every plot across one year (micro_global function). This climatic model takes detailed environmental variables into account, such as elevation, slope, solar radiation and absorptivity[69].

Additionally, for all study plots climate data (BIOCLIM+: mean annual air temperature (bio1); mean daily maximum air temperature of the warmest month (bio5); mean daily minimum air temperature of the coldest month (bio6)) were extracted from CHELSA data base. This dataset is based on temperature measurements of climate station from the years 1981–2010. This was done to have climate data with

higher comparability between the geographic regions and to model future temperatures at the study plots. CHELSA was chosen as it is particularly suitable for modelling climate along mountain slopes due to its terrain-based downscaling of global data (https://chelsa-climate.org). Both bio1 and bio5 climatic variables were highly correlated to estimates of the same measures based on the field-measured climatic data (Extended Data Fig. 9 and Supplementary Information).

Surface temperatures for all study plots were derived from the ECOSTRESS sensor, a radiometer mounted on the International Space Station that measures surface temperatures with a resolution of 70 m (ref. 29). As ECOSTRESS measures surface temperatures at a low temporal resolution and significant parts of the data were filtered due to low data quality (for example, due to cloud cover), surface temperature measurements of study plots were enriched by additional measurements of 50 randomly selected locations within a buffer of 5 km around each plot. For all additional locations the elevational level was determined using the NASA SRTMGL1 digital elevation model (v.003) with a spatial resolution of 1 arc-second (approximately 30 m). We applied a filter (binary '0b00' = best quality retrieval) to only use high-quality surface temperature data. For better visualization, we only plotted temperatures above the 50th percentile of temperature values in Fig. 3.

### Heat coma models

For better estimating the effects of current and future temperatures on tropical insects, we converted the dynamic CT values to static CT values and calculated heat coma times $t_{coma}$ for present and predicted future environmental temperatures[19]. The static CT model includes a thermal sensitivity coefficient $z$, which describes how knockdown time changes with temperature. Across different insect species, $z$ varies between 1 and 5, with an average value of 3 (ref. 19). To provide additional estimations of $z$, we tested ramping rates from 1 to 0.06 °C min$^{-1}$ with three species of leaf cutter ants in the Neotropics (*Acromyrmex coronatus*, *Acromyrmex octospinosus* and *Atta colombica*) as model organisms, since they could be easily found along an elevation gradient from 270–1,349 masl. In the ant data, we found $z$ to range from 2.16–5.50 along the gradient, which overlapped with reports from the literature. For the final static value calculation, we used $z = 3$, following a conservative approach. Under the assumption of higher $z$ values, $t_{coma}$ would be shorter than the values reported in Fig. 3.

For the calculation of the effect of environmental temperatures on $t_{coma}$ of insect communities under current climate and climate change scenarios we extracted community-level $CT_{max}$ values from the lowest plot and applied it across the whole elevational gradient, a rather conservative approach, assuming that the thermal sensitivity of insects across the elevation gradient can, by species turnover, adaptation, or acclimatization, increase to values of $CT_{max}$ currently found in lowland species. We calculated $t_{coma}$ assuming an average $CT_{max}$ (median $CT_{max}$ of all lowland insects), for a 25% quartile of $CT_{max}$ (termed 'more heat-sensitive insects') and for a 10% quantile of $CT_{max}$ ('most heat-sensitive insects'). In the Neotropics, the 10% quantile of all insects measured in the lowland $CT_{max}$ was 40 °C, the 25% quantile was 41 °C, and the median was 42 °C. In the Afrotropics, the 10% quantile of $CT_{max}$ was 43 °C, 45 °C (25% quantile) and 47 °C (median). For future climate projections (2071–2100) under the three shared socio-economic pathways SSP1-2.6 (sustainable scenario), SSP3-7.0 (medium-high scenario) and SSP5-8.5 (high scenario) we used the GFDL-ESM4 climate model predictions (highest priority; https://protocol.isimip.org/#/ISIMIP3b/biodiversity) of bio5 supplied by the CHELSA database[50]. We extracted the mean daily maximum air temperature of the warmest month (bio5) and calculated the anomaly based on the difference between current and future bio5 temperatures (Extended Data Fig. 8). We then estimated the future air temperatures by adding these anomalies to the current environmental temperatures measured in the field (Neotropics) and modelled microclimate (Afrotropics), as well as for the surface temperatures (ECOSTRESS) on each plot. Using this approach, we incorporated the fine-scaled temporal variation in environmental temperatures along the Afrotropical and Neotropical elevation gradients to future climate projections. Thereby, we assume that the differences of environmental temperatures at small temporal scale to bio5 remains the same under future climate projections (details in Supplementary Information). We acknowledge that surface temperatures may not warm at the same rate than air temperatures. The difference between surface and air temperatures is expected to increase with increasing temperatures, particularly in open areas[70]. Thus, our method provides conservative estimates of future surface temperatures. For calculating the percentage of critical temperatures (leading to heat coma within an exposure time of 8 h), we related for each study plot the number of critical temperatures to the total number of temperature values (including 100% of all temperature measurements—that is, including night temperatures).

While reported results are mainly based on the GFDL-ESM4 climate model, all analyses were additionally calculated with a multi-(climate) model ensemble. The detailed methods are described in the Supplementary Information and results of the multi-model-ensemble are shown in Supplementary Fig. 1.

### Inclusion and ethics statement

Fieldwork in Peru and Kenya was conducted in collaboration with local research institutions, including the Universidad Peruana Cayetano Heredia and the Museo de Historia Natural in Lima, and the University of Embu in Kenya. All necessary permits were obtained from the respective national authorities (see Acknowledgements). Local scientists were involved in study design, data collection and manuscript preparation, and are included as co-authors. Field logistics were supported by local assistants, who are acknowledged accordingly. The research supported local capacity building through training initiatives and shared data access. The collection of biodiversity data addresses locally relevant priorities and contributes to future monitoring and conservation efforts.

### Reporting summary

Further information on research design is available in the Nature Portfolio Reporting Summary linked to this article.

### Data availability

All field-collected data are publicly available in the FigShare repository (https://doi.org/10.6084/m9.figshare.28891307 (ref. 71)). Climate data from CHELSA are publicly available at https://www.chelsa-climate.org/ (BIOCLIM+ dataset). ECOSTRESS surface temperatures are available from NASA Earthdata (https://doi.org/10.5067/ECOSTRESS/ECO_L2G_LSTE.002 (ref. 29)). Insect genomes are from InsectBase 2.0 (https://v2.insect-genome.com/).

### Code availability

All code is properly cited within the manuscript and publicly available at Figshare (https://doi.org/10.6084/m9.figshare.28891307 (ref. 71)).

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

**Acknowledgements** We thank the Amazon Conservation Association (ACCA), and J. S. Tintaya for help with fieldwork in Peru. We are grateful to SERNANP for access to Manu National Park (18-2022-SERNANP—JEF), and to SERFOR for helping with and providing research permits in Peru (D001044-2022-MIDAGRI-SERFOR-DGGSPFFS-DGSPFS). We acknowledge the JRS Biodiversity Foundation (60930) for funding and the University of Embu for hosting and logistical support in Kenya. We thank S. König for the identification of Orthoptera specimens. We thank K. Talam, C. Mumo, N. Mkombola and L. Gaithuma for assisting T. Schmitzer in fieldwork in Kenya, authorized by NACOSTI under license no. NACOSTI/P/22/20735. This study was accomplished within the scope of the Research Unit ANDIV (https://www.andiv.biozentrum.uni-wuerzburg.de) and funded by the Deutsche Forschungsgemeinschaft (DFG) under grants PE 1781/4–1, BR 2280/9–1, STE 957/29–1 and KE 1743/12–1.

**Author contributions** M.K.P. conceived the idea for the study. M.K.P. and K.L.H. designed the study. G.B., A.K., M.O., I.S.-D. and M.K.P. acquired funding. K.L.H., T.S., P.A.-A., Y.C.-C. and A.P. arranged logistics with the help of A.F., A.L.-T., F.Y., M.A. and M.O. K.L.H., T.S., A.A. and M.C. collected the data. K.L.H. analysed the data with advice from O.M., M.K. and M.K.P. K.L.H. and M.K.P. wrote the first drafts of the manuscript. All authors revised the manuscript and contributed to the final version.

**Funding** Open access funding provided by Julius-Maximilians-Universität Würzburg.

**Competing interests** The authors declare no competing interests.

**Additional information**
**Correspondence and requests for materials** should be addressed to Kim L. Holzmann.

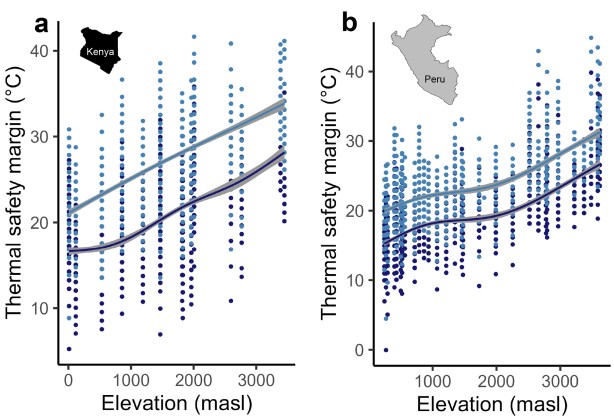

**Extended Data Fig. 1 | Thermal safety margins of tropical insects. a**, Margins along the Afrotropical elevational gradient (Kenya) and **b**, the Neotropical elevational gradient (Peru). Light blue dots are calculated margins based on mean annual temperature (bio1 from CHELSA), dark blue those calculated from the average maximum daily temperature of the warmest month (bio5); where each dot is a field-measured $CT_{max}$ value minus the respective air temperature. Trend lines were calculated with generalized additive models (GAMs) with the smooth term parameter set to k = 5. Afrotropics: edf = 3.797, F = 76.19, p < 2e-16. Neotropics: edf = 3.531, F = 190.8, p < 2e-16. Shaded areas show the 95% CI.

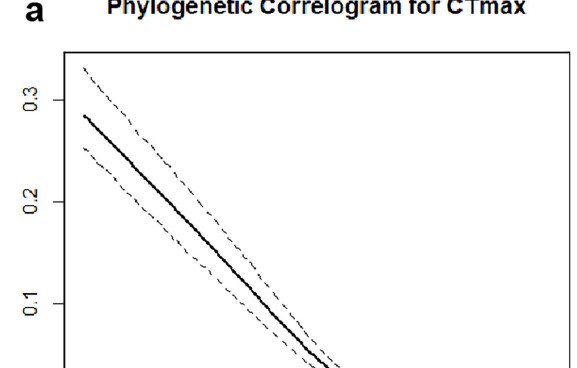

**a** Phylogenetic Correlogram for CTmax

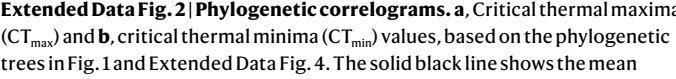

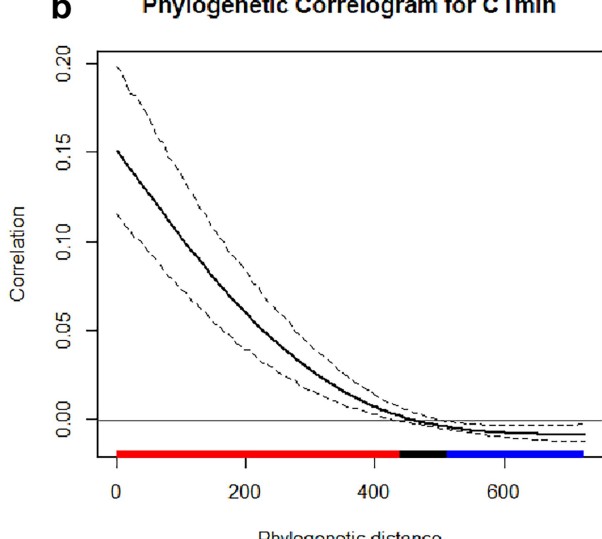

**b** Phylogenetic Correlogram for CTmin

**Extended Data Fig. 2 | Phylogenetic correlograms. a**, Critical thermal maxima ($CT_{max}$) and **b**, critical thermal minima ($CT_{min}$) values, based on the phylogenetic trees in Fig. 1 and Extended Data Fig. 4. The solid black line shows the mean Moran's I autocorrelation values, the dotted lines represent the 95% CI. Red colour along the x-axis indicates a positive significant correlation and blue a negative significant correlation.

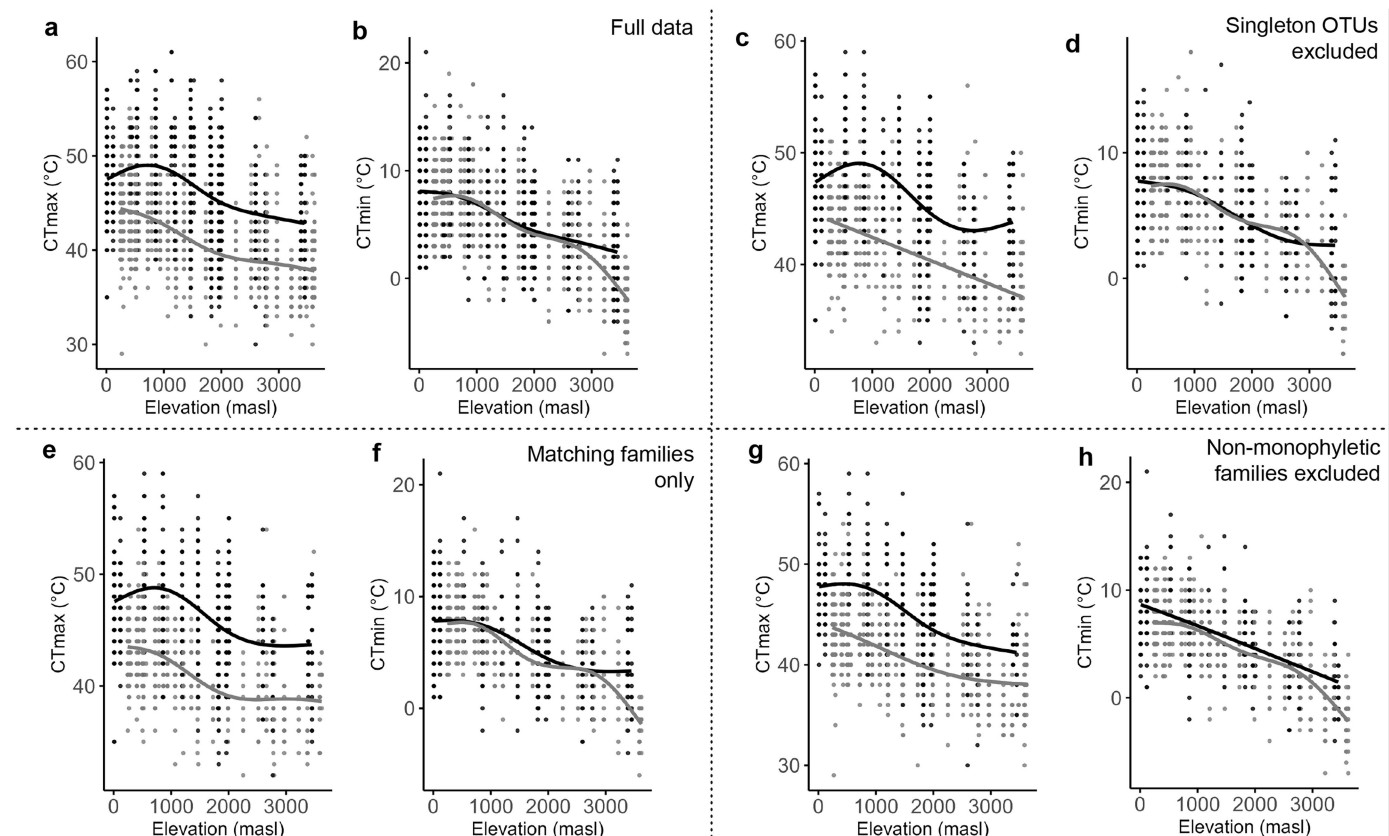

**Extended Data Fig. 3 | Robustness tests.** Elevational trends in critical thermal maxima (CT$_{max}$) and critical thermal minima (CT$_{min}$) for the full data set (**a**,**b**) and data subsets testing modifications of phylogenetic analyses under various scenarios (as described in SI). The data scenarios include the exclusion of (1) species represented by single individuals (**c**,**d**; 'Singleton OTUs excluded'), (2) families that did not match with morphological and barcoding ID (**e**,**f**; 'Matching families only'), and (3) families assumed as non-monophyletic in recent phylogenetic studies of specific insect orders (**g**,**h**; 'Non-monophyletic families excluded'). Shown are trend lines derived from generalized additive models for the data set from Peru (grey) and Kenya (black) with original data values represented by single dots.

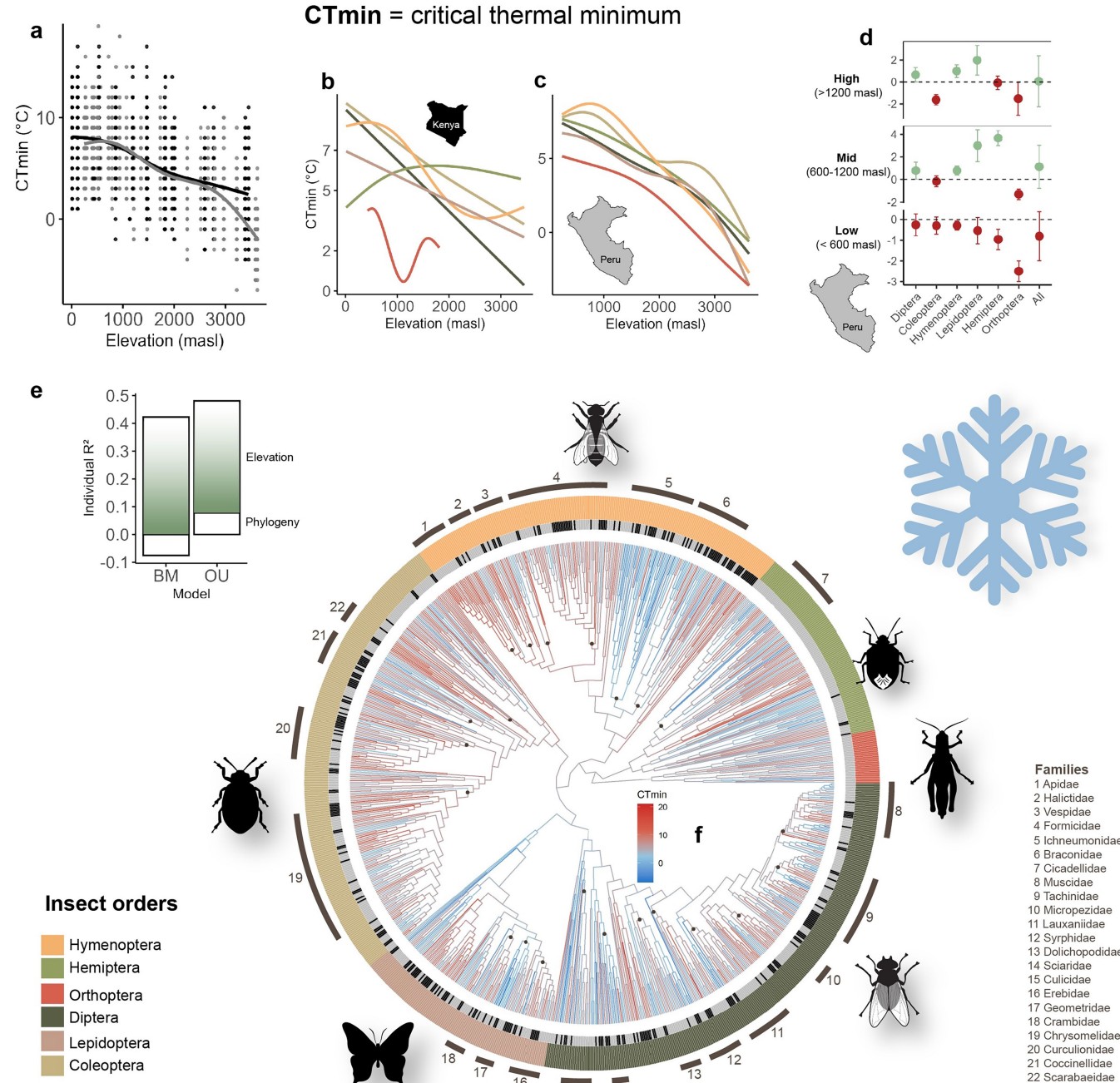

**CTmin = critical thermal minimum**

**Families**
1 Apidae
2 Halictidae
3 Vespidae
4 Formicidae
5 Ichneumonidae
6 Braconidae
7 Cicadellidae
8 Muscidae
9 Tachinidae
10 Micropezidae
11 Lauxaniidae
12 Syrphidae
13 Dolichopodidae
14 Sciaridae
15 Culicidae
16 Erebidae
17 Geometridae
18 Crambidae
19 Chrysomelidae
20 Curculionidae
21 Coccinellidae
22 Scarabaeidae

**Insect orders**
- Hymenoptera
- Hemiptera
- Orthoptera
- Diptera
- Lepidoptera
- Coleoptera

**Extended Data Fig. 4 | Cold tolerance of tropical insects. a**, Field measured critical thermal minima (CT$_{min}$) of insects (n = 3141) decreased along both gradients; **b,c**, a pattern also visible in each major insect order in **b**, the Afrotropical elevational gradient (Kenya) and **c**, the Neotropical elevational gradient (Peru). **d**, Cold hardening was evident after a cold shock experiment (n = 490 insects) at high and mid elevations (green = positive effect), but in the lowlands a sublethal cold shock led to a reduced cold tolerance, i.e. insects not subjected to a cold shock treatment were able to tolerate lower temperatures than those exposed to the cold shock treatment. Dots are means with SE error bars. **e**, The variation in CT$_{min}$ changed with elevation and was less constrained by phylogenetic relatedness than CT$_{max}$. The Ornstein-Uhlenbeck (OU) model had a better fit than the Brownian motion (BM) model. The negative individual contribution of phylogeny r$^2$ indicates suppressor effects (i.e. interacting negatively with elevation). **f**, Phylogenetic supertree constructed from all sampled insects with ancient trait value reconstruction of CT$_{min}$ (n = 1836).

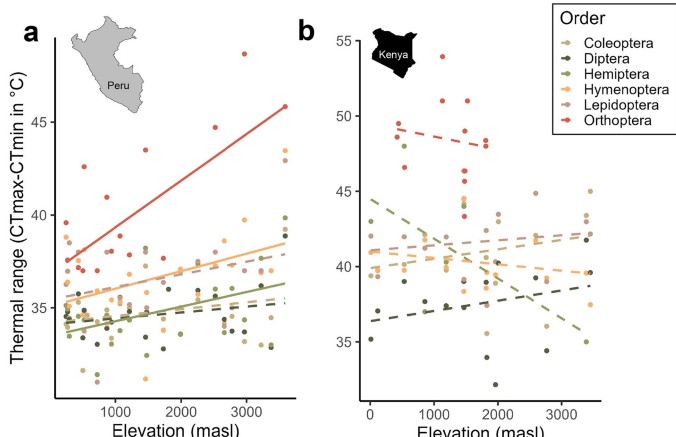

**Extended Data Fig. 5 | Thermal tolerance ranges of insect orders.**
**a**,**b**, Ranges were calculated by subtracting field-measured average $CT_{min}$ from average $CT_{max}$ values along **a**, the Neotropical elevational gradient (Peru) and **b**, the Afrotropical elevational gradient (Kenya). Solid lines indicate significant relations (linear models) and dashed lines non-significant relations.

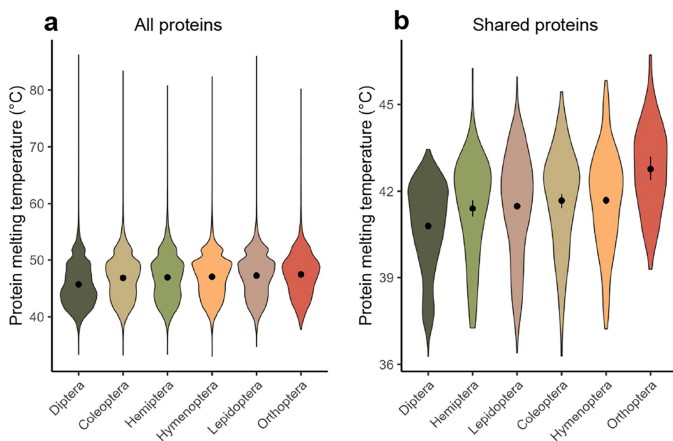

**Extended Data Fig. 6 | Predicted protein melting temperatures. a**, All proteins from high-quality genomes from 354 insect species across six orders (n = 353,435 proteins). The protein melting temperature differs significantly across orders ($F_{5,353429}$ = 2109, p < 2e-16). **b**, Proteins shared among all analysed insect species (n = 2641 proteins). The protein melting temperature differs significantly across orders ($F_{5,2635}$ = 33.5, p < 2e-16). In **b**, only heat sensitive proteins (the 25% most heat sensitive proteins per species) are shown. Black points with error bars are means ± 95% CI and violin polygons show data density. Statistical values are from a linear mixed effect models controlling for species and protein identity, tested with a one-way analysis of variance (two-sided).

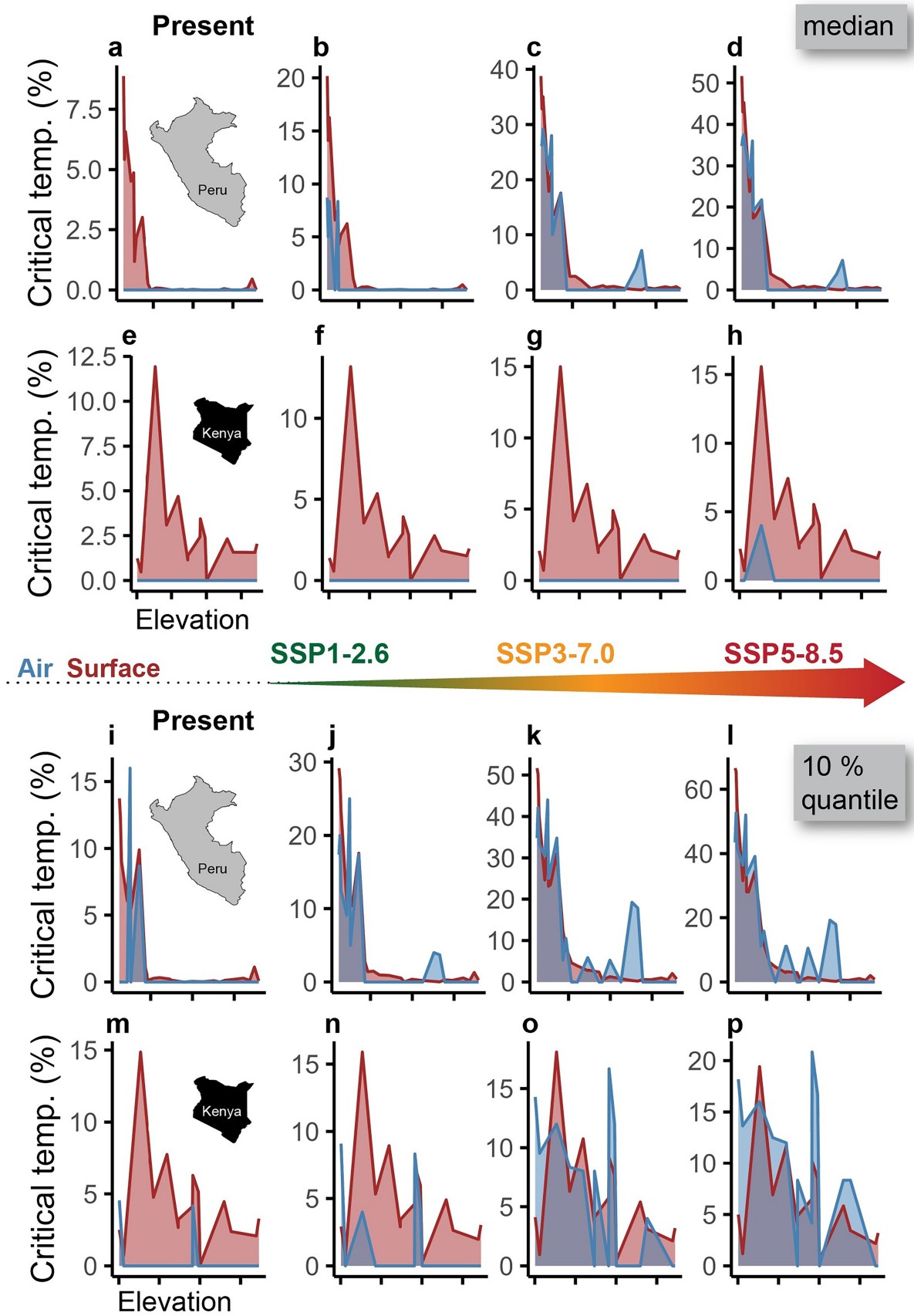

**Extended Data Fig. 7 | Critical temperatures for tropical insects.** Percentage of surface (red) and air (blue) temperatures that are critical (i.e. can cause heat coma in less than 8 h) at the Neotropical (Peru) and East African (Kenya) elevation gradient; (**a**–**h**, upper box) for median $CT_{max}$ values and (**i**–**p**, lower box) for the most heat-sensitive insects (10% quantile of $CT_{max}$). The left graphs (**a**, **e**, **i**, **m**) show critical temperatures for present climatic conditions, the three right graphs in each column show future scenarios considering common climate change projections. Critical temperatures were calculated along the whole gradient assuming the $CT_{max}$ from lowland organisms (see main text and Methods).

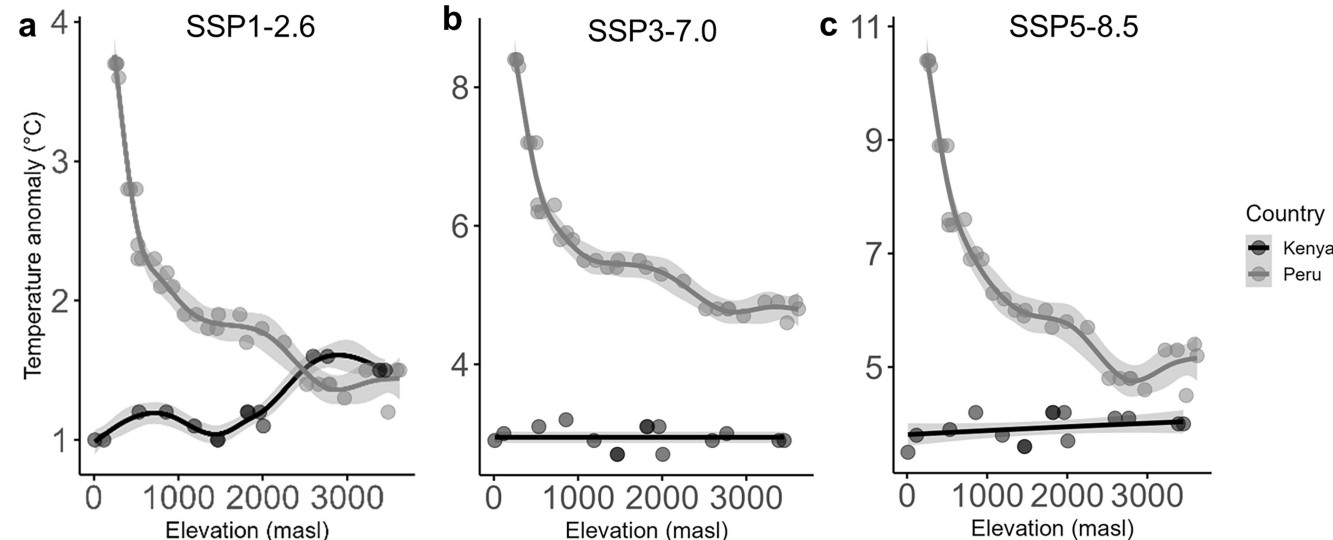

**Extended Data Fig. 8 | Climate change anomalies.** Values for each study plot along the Neotropical (Peru, in grey) and East African (Kenya, in black) elevational gradient were based on the calculated differences between current mean daily maximum air temperature of the warmest month (bio5) and future climate projections (2071–2100) of bio5 under the three shared socio-economic pathways: **a**, SSP1-2.6, **b**, SSP3-7.0, and **c**, SSP5-8.5 using the GFDL-ESM4 climate model[50]. Trend lines were calculated with generalized additive models (GAMs). Shaded area shows the 95% CI.

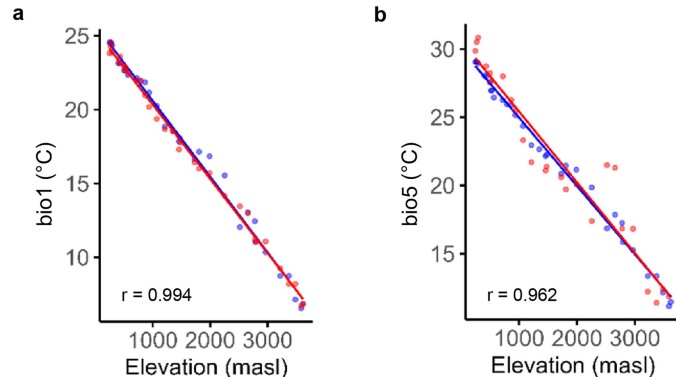

**Extended Data Fig. 9 | Matching of baseline data of CHELSA BIOCLIM+ with field data. a,b**, Trends (lines) and plot level (dots) values of **a**, mean annual temperature measure (bio1) and **b**, the maximum temperature of the warmest month measure (bio5) from CHELSA (blue) and calculated from temperature measurements of TMS loggers placed in the understorey of all study plots (see Methods for more details) (red). Shown are trend lines from ordinary linear models. R values show Pearson correlation coefficients.

**Extended Data Table 1 | Results of phylogenetic linear models of the effect of elevation on (a) critical thermal maxima (CT$_{max}$) and (b) critical thermal minima (CT$_{min}$)**

| Model | Estimate | Std. Error | T value | p value | R² | AIC |
|---|---|---|---|---|---|---|
| **(a) CTmax** | | | | | | |
| BM | -0.00138 | <0.001 | -14.58 | <0.001 | 0.14 | 7113 |
| OU | -0.00155 | <0.001 | -15.96 | <0.001 | 0.17 | 6889 |
| **(b) CTmin** | | | | | | |
| BM | -0.00209 | <0.001 | -24.21 | <0.001 | 0.38 | 4900 |
| OU | -0.00220 | <0.001 | -26.11 | <0.001 | 0.41 | 4677 |

Compared were the Brownian Motion (BM) and Ornstein-Uhlenbeck (OU) model.

# Reporting Summary

## Statistics

For all statistical analyses, confirm that the following items are present in the figure legend, table legend, main text, or Methods section.

| n/a | Confirmed | |
|---|---|---|
| ☐ | ☒ | The exact sample size (*n*) for each experimental group/condition, given as a discrete number and unit of measurement |
| ☐ | ☒ | A statement on whether measurements were taken from distinct samples or whether the same sample was measured repeatedly |
| ☐ | ☒ | The statistical test(s) used AND whether they are one- or two-sided<br>*Only common tests should be described solely by name; describe more complex techniques in the Methods section.* |
| ☐ | ☒ | A description of all covariates tested |
| ☐ | ☒ | A description of any assumptions or corrections, such as tests of normality and adjustment for multiple comparisons |
| ☐ | ☒ | A full description of the statistical parameters including central tendency (e.g. means) or other basic estimates (e.g. regression coefficient) AND variation (e.g. standard deviation) or associated estimates of uncertainty (e.g. confidence intervals) |
| ☐ | ☒ | For null hypothesis testing, the test statistic (e.g. $F$, $t$, $r$) with confidence intervals, effect sizes, degrees of freedom and $P$ value noted<br>*Give P values as exact values whenever suitable.* |
| ☒ | ☐ | For Bayesian analysis, information on the choice of priors and Markov chain Monte Carlo settings |
| ☒ | ☐ | For hierarchical and complex designs, identification of the appropriate level for tests and full reporting of outcomes |
| ☐ | ☒ | Estimates of effect sizes (e.g. Cohen's *d*, Pearson's *r*), indicating how they were calculated |

*Our web collection on statistics for biologists contains articles on many of the points above.*

## Software and code

Policy information about availability of computer code

| Data collection | N/A |
|---|---|
| Data analysis | All analyses were conducted in R (version 4.3) with the following packages: DECIPHER v2.30.0, ape v5.8, phylocomr v0.3.4, ips v0.0.12, phytools v2.3.0, ggtree v3.10.1, ggplot2 v3.5.1, ggnewscale v0.5.0, phylosignal v1.3.1, phylolm v2.6.5, geiger v2.0.11, phylolm.hp v0.0-3, NicheMapR v3.3.2, lme4 v1.1-36, DHARMa v0.4.7, officer v0.6.7, dplyr v1.1.4, tidyR v1.3.1, readxl v1.4.3, openxlsx v4.2.7.1. We used Anaconda PowerShell and Python (v. 3.13) to run DeepSTABp. The REfined Single Linkage alogorithm was used from BOLD version 4 (https://v4.boldsystems.org/). All code is available at https://doi.org/10.6084/m9.figshare.28891307. |

For manuscripts utilizing custom algorithms or software that are central to the research but not yet described in published literature, software must be made available to editors and reviewers. We strongly encourage code deposition in a community repository (e.g. GitHub). See the Nature Portfolio guidelines for submitting code & software for further information.

## Data

Policy information about availability of data

All manuscripts must include a data availability statement. This statement should provide the following information, where applicable:
- Accession codes, unique identifiers, or web links for publicly available datasets
- A description of any restrictions on data availability
- For clinical datasets or third party data, please ensure that the statement adheres to our policy

All field-collected data is publicly available in the FigShare repository (https://doi.org/10.6084/m9.figshare.28891307). Climate data from CHELSA is publicly available at https://www.chelsa-climate.org/ (BIOCLIM+ data set). ECOSTRESS surface temperatures are available from NASA Earthdata (https://doi.org/10.5067/ECOSTRESS/ECO_L2G_LSTE.002 [29]). Insect genomes are from InsectBase 2.0 (https://v2.insect-genome.com/).

## Research involving human participants, their data, or biological material

Policy information about studies with human participants or human data. See also policy information about sex, gender (identity/presentation), and sexual orientation and race, ethnicity and racism.

| | |
|---|---|
| Reporting on sex and gender | N/A |
| Reporting on race, ethnicity, or other socially relevant groupings | N/A |
| Population characteristics | N/A |
| Recruitment | N/A |
| Ethics oversight | N/A |

Note that full information on the approval of the study protocol must also be provided in the manuscript.

# Field-specific reporting

Please select the one below that is the best fit for your research. If you are not sure, read the appropriate sections before making your selection.

☐ Life sciences  ☐ Behavioural & social sciences  ☒ Ecological, evolutionary & environmental sciences

For a reference copy of the document with all sections, see nature.com/documents/nr-reporting-summary-flat.pdf

# Ecological, evolutionary & environmental sciences study design

All studies must disclose on these points even when the disclosure is negative.

| | |
|---|---|
| Study description | In the study we tested the effects of climate (mean annual temperature, quantitative variable) and phylogenetic relations on thermal limits of tropical insects along two elevation gradients in the Andean-Amazonian ecosystem (Peru) and in East Africa (Kenya). We tested the underlying mechanism of physiological thermal limits by investigating insect protein melting temperatures. For most response variables the sample size was 26 in the Neotropics and 15 in the Afrotropics; for few response variables the sample size was lower as data was missing for some study sites. The sample size is indicated for each response variable in the method section. As all study sites were geographically seperated by an elevational interval of 250 m and a linear distance of at least 400 m, and on average all study site pairs were seperated by 1.5 km, we considered study sites as independent replicates and used gernalized additive or linear models (e.g. linear regression) for nearly all of the analyses. |
| Research sample | We tested thermal limits of insect species across six major orders: Diptera, Coleoptera, Hymenoptera, Hemiptera, Lepidoptera, Orthoptera. |
| Sampling strategy | The number of study sites (26 in the Neotropics, 15 in the Afrotropics) represents a balance between good statistical power and feasability of field work and was chosen based on the great experience of many included authors in the performance of tropical field work and the analyses of ecological data sets. |
| Data collection | Several of the authors (KLH, TS, AA, MC) collected the data in the field. |
| Timing and spatial scale | Data was collected from September 2022 to December 2023 with a gap (January 2023 to April 2023) during the strong rain season in the Neotropics, since roads are often not accessible and field work would be not feasible during this time. |
| Data exclusions | No data was excluded. |
| Reproducibility | We used a common protocol to test thermal limits of insects and measured ~8000 individuals. All details are provided in the |

| Reproducibility | methods, so that our protocol can be repeated. In both geographic regions we applied the same standardized measurements so that data can be compared. We controlled for differences in taxonomy by using DNA sequences from each tested insect to create a phylogenetic tree. |
| --- | --- |
| Randomization | Insect samples were randomly collected at each plot and randomly assigned to be tested either for their upper or lower thermal limit. |
| Blinding | Blinding is not relevant to our study, since we analysed a real mountain ecosystem by standardised methods, that creates a numerical outcome which is not subject to personal interpretation. |

Did the study involve field work?  ☒ Yes  ☐ No

# Field work, collection and transport

| Field conditions | In Peru, the study was carried out along an elevational gradient from 245 meters above sea level (masl) to the tree line at 3588 masl in the Andes of south-east Peru (Kosñipata valley), with continuous and mostly undisturbed wet rainforest/cloud forest. Mean annual temperatures range from 24.3 °C in the lowlands to 6.7 °C at 3600 masl. Mean annual precipitation levels are high with > 1500 mm per year along the whole gradient, peaking at around 1500 masl with ~5000 mm. In Kenya, the study was carried out along an elevational gradient from 11 masl at Watamu to 3450 masl at Mount Kenya including forests, woodland, scrub, and grassland in natural and semi-natural habitats. Mean annual temperatures range from 26.2 °C at the lowest plot to 8.9 °C at the highest plot. The forested parts of the Taita Hills region and Mount Kenya from ~1300–2500 masl are characterized by a tropical montane forest climate with generally high humidity and constant high precipitation (> 1500 mm). |
| --- | --- |
| Location | In Peru, the gradient ranged from -12,57079 -70,09245 at an elevation of 245 masl to -13,09608 -71,62952 at an elevation of 3588 masl. In Kenya, from -3,37667 39,98829 at an elevation of 11 masl to -3,37667 39,98829 at an elevation of 3450 masl. |
| Access & import/export | All data collection were conducted under appropriate permits. In Peru, SERNANP provided access to Manu National Park (N° 18-2022-SERNANP - JEF), and ERFOR provided research permits (Nº D001044-2022-MIDAGRI-SERFOR-DGGSPFFS-DGSPFS). In Kenya, we had permits from the JRS Biodiversity Foundation (grant No: 60930) and in fieldwork was authorized by NACOSTI under License No: NACOSTI/P/22/20735. This study was accomplished within the scope of the Research Unit ANDIV (www.andiv.biozentrum.uni-wuerzburg.de) and funded by the Deutsche Forschungsgemeinschaft (DFG) under grant PE 1781/4-1. |
| Disturbance | We did not experimentally modify the study plots. Insects had to be collected, killed and preserved in order to conduct the study; however, we are confident that this has no impact on the species populations. |

# Reporting for specific materials, systems and methods

We require information from authors about some types of materials, experimental systems and methods used in many studies. Here, indicate whether each material, system or method listed is relevant to your study. If you are not sure if a list item applies to your research, read the appropriate section before selecting a response.

## Materials & experimental systems

| n/a | Involved in the study |
| --- | --- |
| ☒ | ☐ Antibodies |
| ☒ | ☐ Eukaryotic cell lines |
| ☒ | ☐ Palaeontology and archaeology |
| ☐ | ☒ Animals and other organisms |
| ☒ | ☐ Clinical data |
| ☒ | ☐ Dual use research of concern |
| ☒ | ☐ Plants |

## Methods

| n/a | Involved in the study |
| --- | --- |
| ☒ | ☐ ChIP-seq |
| ☒ | ☐ Flow cytometry |
| ☒ | ☐ MRI-based neuroimaging |

# Animals and other research organisms

Policy information about studies involving animals; ARRIVE guidelines recommended for reporting animal research, and Sex and Gender in Research

| Laboratory animals | No laboratory animals were used. |
| --- | --- |
| Wild animals | During the study we collected 7994 insects from six orders (Diptera, Hymenoptera, Coleoptera, Lepidoptera, Orthoptera), comprising 2330 unique genetic units, no morphological identification on species level was made and no information on age, strain or sex were collected. The specimen are preserved and will be stored in museum collections in Peru and Kenya. |
| Reporting on sex | No information on sex has been collected. |
| Field-collected samples | Samples collected in the field were transported back to the station in 2-50 ml tubes equipped with sugar water within two hours, |

| Field-collected samples | maintained at the ambient temperature of the collection site, no exposure to light, and experiments were conducted the same day. Afterwards samples were stored in ethanol. |
| Ethics oversight | No ethical approval was required since lower invertebrates (insects) were tested. |

Note that full information on the approval of the study protocol must also be provided in the manuscript.

## Plants

| Seed stocks | *Report on the source of all seed stocks or other plant material used. If applicable, state the seed stock centre and catalogue number. If plant specimens were collected from the field, describe the collection location, date and sampling procedures.* |
| Novel plant genotypes | *Describe the methods by which all novel plant genotypes were produced. This includes those generated by transgenic approaches, gene editing, chemical/radiation-based mutagenesis and hybridization. For transgenic lines, describe the transformation method, the number of independent lines analyzed and the generation upon which experiments were performed. For gene-edited lines, describe the editor used, the endogenous sequence targeted for editing, the targeting guide RNA sequence (if applicable) and how the editor was applied.* |
| Authentication | *Describe any authentication procedures for each seed stock used or novel genotype generated. Describe any experiments used to assess the effect of a mutation and, where applicable, how potential secondary effects (e.g. second site T-DNA insertions, mosiacism, off-target gene editing) were examined.* |

