## [Peer Review file · Nature]

Limited thermal tolerance in tropical insects and its genomic signature

Corresponding Author: Dr Kim Holzmann

Version 0:

Reviewer comments:

Referee #1

(Remarks to the Author)

A. Summary of the key results. In the manuscript "Limited thermal tolerance in tropical insects and its genomic signature", the authors explore how thermal tolerance changes with elevation along two elevational gradients in Peru and Kenya. The study reports critical thermal maxima and minima for more than 2000 "species" identified using the DNA barcode CO1. Using CO1 sequences, the authors generated a phylogeny to test for phylogenetic signal in CTmax among clades.

Experiments exposing insects to extreme temperatures (shock) and then to increasing temperatures aimed to determine plasticity in thermal tolerance among species. Using published databases, the study includes thermal responses of heat shock proteins in various insects. The study includes a projection of how future temperatures may generate thermal coma.

B. Originality and significance: The study includes an outstanding number of species along two elevational gradients. The main result, that CTmax decreases with elevation in tropical insect species was previously published. The phylogenetic analysis, heat shock protein analyses, and the extinction risk forecasting analysis are interesting.

C. Data & methods: The authors delimited species using the DNA barcode CO1, using sequences with "more than 100 bp". This is problematic. If many of the included sequences are short (i.e. close to 100 - 300 bp) it is possible that this will inflate the number of MOTUS (i.e. identify as another species individuals with sequences too short to match sequences close to ~600 bp).

The authors must be cautious with their interpretation of the phylogenetic analysis. Using DNA barcodes to reconstruct ancestral characters such as thermal tolerance is incorrect. Even with more extensive molecular information per species, it is problematic that the sampling of species and families amongst orders is limited to a few taxa along elevational gradients. The supplementary material should include more details on the number of individuals per OTU. If the sampling was balanced, I would expect about 5 individuals per "species". However, it is very likely that OTUs represented by a single individual were common. The evenness of the sampling should be reported in the manuscript, as well as in the metadata and datasets associated with the paper. Analyses must include OTU as a factor to ensure that the results are not skewed by abundant species. If the authors included CTmax estimates based on a single individual, I think these data should be removed.

The "protein stability predictor" analysis should be part of another manuscript. Data for these analyses were obtained online, not from the sampled insects. The connection between the arguments of thermal tolerance of different proteins and the study performed in the field is not clear.

Using CTmax to model organismal responses to changing temperatures is problematic. CTmax is not fitness, and populations will decline at temperatures lower than those at which insects collapse in the laboratory. Using such extreme temperatures (e.g., heat coma temperatures) to project responses to warming may be misleading, and underestimates the effect of temperature on insect extinctions.

Some minor comments

P4L61. The goal of the comparison with arctic species is not clear.

P4L62. The mechanistic explanation for this statement is not clear.

References. I noticed that some citations do not support the statements included in the manuscript. For example, use books and literature reviews to support very particular statements. Please use primary literature (e.g. refs 13, 17). The text identifies studies as "meta-analyses when" they are not (e.g., refs 7,8,9).

Referee #2

(Remarks to the Author)

Holzmann and colleagues seek to understand how the thermal maxima (CTmax) of insects has evolved across the insect tree of life and its evolutionary potential under climate changes. They found that the evolutionary history of insects rather than local temperature conditions have a stronger influence on the evolution of CTmax. In other words, the evolution of CTmax is constrained by the phylogenetic relatedness and closely related organisms show similar CTmax values. The authors argue that this is supported by the strong phylogenetic signal in CTmax and ancestral state reconstruction for CTmax.

Similar findings have been reported previously (e.g., <https://doi.org/10.1111/j.1365-2435.2012.02036.x> and <https://doi.org/10.1038/s41467-022-32953-2>), but were supported by limited data size or studies not specifically focusing on insects (e.g., <https://doi.org/10.1038/s41467-021-21263-8>). In this study, Holzmann and colleagues addressed the problem by focusing on six insect orders.

I was asked to focus my evaluation on phylogenetic reconstruction. In this regard, the manuscript does show some caveats in methodology. The major concern here is the lack of robustness in the tree topology and relevant evolutionary time. The authors will need to either improve their phylogenetic reconstruction or demonstrate why the current methodology won't affect their conclusions. I will elaborate this in more details:

In a nutshell, the authors adopted a family-level tree from a published work and added COI/NU trees to each of the pre-defined family nodes. I have a few major concerns about this methodology:

1. The adopted "family root tree" is far from being robust. Although the Rainford et al (2014) tree had an impressive 82.1% family converge for extant insects, this tree was built on just a few gene markers, which has been shown less effective by recent phylogenomics studies when compared with genome-level markers, especially when the focus is placed at deep-level relationships. Perhaps an even larger concern is that the Rainford et al phylogeny was based on a chimeric dataset, i.e., the representative data matrix for a given family was "assembled from multiple individuals and species" to improve gene coverage among taxa. An in-depth discussion on the 2014 tree is clearly out of scope here. But the reason for comfortably adopting this tree for the present study needs to be justified. In fact, family-level relationships of many insect orders have been re-examined heavily in the past 10 years using large-scale genomic data, for example, Coleoptera (<https://doi.org/10.1098/rso.211771>), Orthoptera (<https://doi.org/10.1038/s41467-020-18739-4>), Lepidoptera (<https://doi.org/10.1038/s41559-023-02041-9>), Hymenoptera (<https://doi.org/10.1038/s41467-023-36868-4>). The authors should assess how the family-level insect phylogeny from Rainford et al (2014) differs from these latest studies and the potential influence on their conclusion.
2. The family assignments for newly collected insects need to be carefully validated. Here, the authors focus on six insect orders, Coleoptera, Diptera, Hymenoptera, Lepidoptera, Hemiptera and Orthoptera. When possible, the authors assigned individual insects (collected from Kenya and Peru) to order- or family-level taxonomy based on morphology, before the specimens were sequenced for COI barcodes. Afterwards, the barcodes were clustered into OTUs, where the COI sequences were compared against the BOLD system for a family-level assignment. The authors employed an arbitrary BOLD similarity cutoff (>=80%) for this purpose (Line 403). Unfortunately, family assignment based on COI barcodes is known to be less effective when compared to those of lower taxonomic hierarchies (genus, species), especially when closely related species was not present in the reference library. In this case, it is not uncommon that OTUs can be assigned to wrong families. I would argue that mistakes in family placement will have a significant impact on the interpretation of trait evolution, which is even worse than using a questionable phylogeny. As a general practice, the family level identification needs to be supported by morphology. If a consensus is indeed reached between morphology and barcode ID, this needs to be clarified and a summary of the results showing the congruence/discrepancy in this process should be presented.
3. Discrepancies in family coverage. I believe we can safely assume that the family coverage between the adopted family tree and the barcoded insects is not identical. However, it is not clear how the authors dealt with this. What happened when (if) the new samples contained families that were not included in the 2014 phylogeny, or vice versa? Does this discrepancy change the results and conclusions on CTmax evolution?
4. Monophyly of pre-defined families and barcode sub-trees. Families included in the 2014 phylogeny are presumed monophyletic in the present study. However, taxonomic revision, including those at the family-level, is expected to be a persistent endeavor. It is inappropriate to intuitively assume that the family definition remains unchanged over the past decade. In cases where a pre-defined family is non-monophyletic, the corresponding node should not serve as an anchor point to which the barcode tree is to be attached. Another relevant issue is the use of the NJ method in sub-tree construction. This is problematic at least from a phylogenetics perspective. A character-based method, for example a maximum likelihood tree should instead be applied here. In addition, any inferred tree should be provided with support values to show how reliable the tree is (e.g., bootstrap values for the nodes).
5. A relevant question is the estimation on branch length of the new tree that integrated the "root" topology from Rainford et al (2014) and "leaves" from newly collected insects. In a phylogenetic comparative analysis including ancestral state reconstruction, a time tree is usually used as the input. The robustness of the topology and branch length of the input tree is important as many trait evolution models (e.g., the Brownian Motion model) attempt to model trait state (or value) changes over time (or proportional to time). An appropriate comparative method should recognize uncertainties associated with both topology and branch lengths, and subsequently incorporate these considerations into the analysis. (page41, <https://lukeharmon.github.io/pcmipdf/phylogeneticComparativeMethods.pdf>; <https://doi.org/10.1101/2025.06.16.659089>). In this study, "the Bldfj function from phylocom... estimates node ages based on fossil calibration points while unknown ages

Referee #5 (Remarks to the Author):

I am satisfied with the response to my specific comments as well as with the revised manuscript as a whole.

KLJH: We want to thank referee 5 for their effort in reviewing our manuscript.

Referee #6 (Remarks to the Author):

Holzmann et al. explore the evolutionary patterns of upper thermal tolerance (CTmax) across insects and consider the extent to which these traits may respond to climate warming. Their analyses indicate that CTmax variation is shaped more strongly by shared evolutionary history than by present day thermal conditions. This pattern implies a considerable degree of phylogenetic conservatism, with closely related lineages exhibiting comparable upper thermal limits. The authors substantiate this inference through the detection of a pronounced phylogenetic signal in CTmax and by reconstructing ancestral thermal states. My review was requested to concentrate specifically on the thermal stability prediction and its evaluation. From this standpoint, I identify a methodological limitation that, in my view, does not alter the main conclusions. The tool used by the authors, DeepSTABp, is well suited for predicting distributions of protein thermal stability based on structural properties. However, when applied across species, additional factors, such as differences in compatible solutes or other aspects of the cellular protein environment, may influence protein stability. DeepSTABp attempts to account for such effects by incorporating optimal growth temperature as an additional parameter.

In the present study, the authors use the same temperature for all predictions. Because their analysis focuses on differences in the lower quartile of the predicted stability distributions, the observed effects remain valid. Moreover, their comparative analyses are most likely robust to this assumption due to aggregation, which I find to be a thoughtful and effective analytical choice. While the absolute temperature values on the y axis may be shifted in a more or less linear fashion, such a shift would not affect the qualitative pattern or the central findings of the study.

Therefore, I agree with the authors' interpretation and consider their conclusions to be well supported despite the noted methodological limitation.

KLJH: We thank reviewer 6 for their thoughtful assessment of the thermal stability prediction component. We agree that while DeepSTABp has its methodological limitations, they will not affect our qualitative results across a broad insect community. Indeed, all comparative analyses provided highly robust results.

are evenly distributed between known nodes" (Lines 411–412), which may introduce inaccurate estimation of branch lengths. The potential effect of this uncertainty in branch length on ancestral state reconstruction should be assessed.

6. Incomplete taxon sampling can also affect the outcomes of comparative analyses, such as ancestral state reconstruction. One approach to address this is to sequentially introduce hypothetical missing taxa into the phylogenetic tree and re-run the analyses on these expanded phylogenies to evaluate how ancestral state reconstruction is impacted (e.g., <https://doi.org/10.1073/pnas.0811421106>). Alternatively, researchers can randomly remove taxa from various clades—

particularly those that are well-sampled—and examine how this reduction in taxon sampling influences the inference of ancestral states (e.g., <https://doi.org/10.1111/jbi.12211>).

7. Finally, the authors tried to understand the evolution of the trait C_{max} using the OU and BM models. More specifically, the authors tried to test if this trait evolves randomly or is under stabilizing selection towards an optimum. However, there are many trait evolution models (e.g., BM, OU, EB, lambda, rate, trend, delta and white) available and the rationale behind the model choice needs to be justified. This can be done with the fitContinuous() function in the package geiger. Ancestral state reconstruction should be carried out with the best-fit model.

Minor issues:

Line 102: When assessing the amount of phylogenetic signal in the trait C_{max}, the authors tested only Pagel's λ but not Blomberg et al.'s (2003) K, without providing corresponding p-values and simulation numbers (should be $\geq 10,000$). Please clarify.

Line 403: The application of a cutoff $\geq 80\%$ for family assignment needs to be justified or backed up by a reference.

Line 406: Reference 48 should be updated. The manuscript has been published.

Referee #3

(Remarks to the Author)

I co-reviewed this manuscript with one of the reviewers who provided the listed reports.

Referee #4

(Remarks to the Author)

This study presents a compelling integration of experimental thermal tolerance data with elevation-based ecological fieldwork and climate projections, offering a novel framework to evaluate tropical insect vulnerability to future warming. The authors combine empirical measurements of critical thermal maxima with modeled exposure times under current and projected microclimatic conditions. I would suggest that this study is published after the points with respect to the climate projections I raise below are appropriately dealt with.

Originality and Significance:

The manuscript brings together physiological data, environmental microclimate modeling, and biodiversity-relevant climate scenarios in a way that is innovative and timely. To my knowledge, few studies integrate insect thermal physiology with high-resolution, elevation-based climate exposure modeling at this scale. The study addresses a pressing and underexplored question, how tropical insect communities may be affected by climate change across elevational gradients, and does so with a creative and potentially impactful framework in my opinion.

Data & Methodology: Validity and Quality:

While the experimental and field components of the study are well executed and clearly presented, the climate projection component requires revision. The authors apply long-term mean climate anomalies (Abio5 from CHELSA, based on 1981–2010 and 2071–2100 climatologies) directly to high-resolution microclimate data collected over a single year (2022–2023). This assumes that the observed year is representative of the historical climatological baseline, which is unlikely given interannual climate variability and recent warming trends (e.g., El Niño). As a result, the projected future temperatures may be biased high or low depending on whether the sampling year was anomalously warm or cool. Additionally, while the delta-change method may be appropriate for below-canopy air temperatures, the same anomalies are also applied to land surface temperatures derived from ECOSTRESS. This assumes that surface temperatures — representing radiative skin temperatures — warm at the same rate as air temperatures, which may not be valid in forested environments due to differential radiation balance, evapotranspiration, and canopy feedbacks. These assumptions should be explicitly acknowledged and discussed.

Main point: Thermal sensitivity under climate change

The authors apply long-term mean climate anomalies (Abio5 from CHELSA, based on 1981–2010 and 2071–2100 climatologies) directly to high-resolution microclimate data collected over a single year (2022–2023). This assumes that the observed year is representative of the historical climatological baseline, which is unlikely given recent global warming trends and interannual variability (e.g., El Niño). As a result, the projected future temperatures may be biased high or low depending on whether the sampling year was anomalously warm or cool.

A more robust approach would be to baseline-correct the observed microclimate data to the 1981–2010 mean before applying the climate anomaly. While the authors acknowledge limited long-term microclimate data availability, explicitly addressing this assumption and its potential effects on future temperature estimates would strengthen the methodological

Neotropics). Re-running analyses using morphology-based assignments resulted in only ~2% average change in order-level C_{Tmax} estimates, with the elevation trend preserved.

3. Families present in the current dataset but not in Ramford et al. (2014) were previously inserted adjacent to sister families. Reanalysis excluding these manually placed families produced an average change of 0.04% in C_{Tmax} estimates.

4. The use of a maximum-likelihood tree with bootstrap support, instead of the earlier neighbor-joining tree, provides a more robust phylogenetic framework.

5. The authors introduced branch-length perturbations (10% jitter) across 50 replicate trees to evaluate variation in ancestral-state reconstruction. We would appreciate clarification on the choice of the 10% perturbation level and whether alternative magnitudes (e.g., 5%, 20–50%) were considered or tested.

6. Randomly removing 50% of taxa in 100 replicate analyses produced C_{Tmax} estimates congruent with the full dataset at both order- and family-level nodes.

Overall, we find that the additional analyses substantially strengthen the study and address the primary concerns we raised. The conclusions now appear more robust to taxonomic assignment differences, phylogenetic uncertainty, and sampling variation.

Xin Zhou and Guanliang Meng

KLH: We want to thank Xin Zhou and Guanliang Meng for their valuable review, with a particular focus on the phylogenetic analyses. Regarding the fifth point, they asked for a justification of the 10% perturbation level. Indeed, we had initially also tested both smaller and larger jitter magnitudes, but this resulted in highly similar means with proportional smaller (e.g. 5% or larger (50%) variations. We decided to apply a $\pm 10\%$ proportional perturbation of branch lengths to represent moderate divergence-time uncertainty typical of backbone-constrained and fossil-calibrated phylogenies. Such proportional jittering is commonly used in sensitivity analyses of phylogenetic comparative methods to evaluate robustness without introducing biologically unrealistic temporal variation. We are confident that moderate variation in branch lengths will not affect our main conclusions.

Referee #3 (Remarks to the Author):

I co-reviewed this manuscript with one of the reviewers who provided the listed reports.

Referee #4 (Remarks to the Author):

The authors have addressed all my concerns where possible and the inclusion of a multi model ensemble for climate change projections now delivers the required uncertainty assessment with respect to climate change too. I do not have further concerns regarding this study.

KLH: We want to thank referee 4 for their thorough review including helpful recommendations regarding the climate change projections.

transparency of the study.

The authors also apply a delta-change approach using derived air temperature anomalies (bio5) to project future environmental temperatures. While this method is potentially appropriate for below-canopy air temperatures, the same anomalies are also applied to land surface temperatures derived from ECOSTRESS. This assumes that surface temperatures, representing radiative skin temperature of the land or canopy, will increase at the same rate as air temperature, which may not hold true in forested environments where evapotranspiration, canopy structure, and radiation balance can modulate surface warming differently. I recommend the authors explicitly acknowledge this assumption and its limitations, particularly given the distinct physical processes governing air vs. surface temperature responses to climate change. Clarifying this in the methods and discussion would improve transparency and help readers interpret the results more cautiously.

To improve the accuracy of future below-canopy air temperature projections, I suggest an alternative to the current delta-change method. Specifically, the authors could calculate the difference between open-air macroclimate temperature (e.g., CHELSA bio5 for 2022–2023 or local weather station data) and the observed below-canopy air temperatures during the same period. This would yield an empirical distribution or average offset that quantifies local microclimate buffering under current conditions. This offset could then be applied to CHELSA's historical (1981–2010) and future (2071–2100) bio5 values, thereby generating below-canopy temperature estimates that are both baseline-aligned and regionally adjusted for canopy effects. This would reduce the risk of bias due to interannual anomalies in the observed data and better reflect how microclimates are expected to warm relative to regional climate. Since the authors also have daily measurements of climate data, they could apply a quantile mapping rather than a simple delta change, accounting for the fact that air-temperature and below canopy temperatures might not always be highly correlated at temperature extremes, or nonlinear.

Appropriate Use of Statistics and Treatment of Uncertainties:

Statistical treatment of thermal physiology and model fitting appears appropriate. However, uncertainty in future temperature projections — particularly regarding the assumption that a single year of microclimate data can serve as a baseline for delta-change — should be more clearly addressed. Treatment of variance, extremes, and potential non-linear effects in warming scenarios could be improved.

Conclusions: Robustness and Validity:

The overall conclusions are well supported by the experimental physiology and elevational comparison. However, the credibility of the future exposure analyses depends on addressing the climate projection concerns noted above. Until those are resolved, the broader ecological conclusions about future vulnerability remain tentative.

Suggested Improvements:

- 1) To better align microclimate data with the CHELSA climatology baseline, I recommend estimating the difference between open-air macroclimate (e.g., CHELSA or local weather station data) and below-canopy air temperature during the observed year. This offset can then be applied to both historical and future CHELSA values to generate baseline-aligned below-canopy projections.
- 2) Given the availability of measured daily temperature data, the authors could implement a quantile mapping approach rather than a simple additive delta-change. This would account for potential non-linearity in the relationship between air temperature and below-canopy microclimate, particularly at extremes.
- 3) Clarify in the methods that the surface temperature data (from ECOSTRESS) reflect land surface (skin) temperatures and not air temperatures, and justify the assumption that these warm at the same rate as CHELSA air temperature anomalies.
- 4) The authors rely on a single global climate model (GFDL-ESM4) from the CMIP6 archive, following the ISMIP3b protocol, to calculate future climate anomalies (Δ bio5). While this model is widely used and part of an established impact protocol, relying on a single GCM does not capture the full range of uncertainty inherent in future climate projections. Inter-model variability, particularly at regional and elevational scales, can significantly influence ecological forecasts. I recommend that the authors incorporate an ensemble of CMIP6 models, such as those available through the CHELSA multi-model ensemble.

Clarity and Context:

The manuscript is generally well written and well structured. The abstract, introduction, and conclusions clearly state the motivation and findings. However, the climate projection methods, particularly how future temperatures are generated, require greater detail and clarity.

Referee #5

(Remarks to the Author)

This is the most interesting, comprehensive and innovative manuscript I have seen in the recently emerging wave of studies on thermal tolerance, especially in the tropics. This research topic is a response to the increasing concern about climate change and, in particular, insect attrition in lowland tropical areas, which was raised as a highly controversial possibility almost 20 years ago by Colwell et al. (2008) (although the present paper does not cite this study in this context). This paper is the clearest indication yet that Colwell et al. may be right and that tropical lowland insects may be at serious risk.

The reason why this paper stands out from other similar studies is its comprehensive dataset, which includes more than 2,000 insect species along two long tropical gradients on two continents. In addition, the question of plasticity of T_{max} is partly answered by measuring its changes after heat shock, presumably following upregulation of heat shock proteins. It also

Point-by-point Response

Referees' comments are shown in light grey.

Referee #1 (Remarks to the Author):

Dear editors,

I am very happy with the detailed response to the comments. The study is original and involves many insect species in a broad geographic range. Data and methodology are valid, and the authors ensured to explain the potential issues of phylogenetic analyses using DNA barcodes. The new version of the manuscript is clear and references are adequate. My recommendation is to accept in its current form.

Carlos Garcia-Robledo, Ph.D.

Associate Professor

Department of Ecology and Evolutionary Biology

University of Connecticut

KLH: We want to thank Dr. Garcia-Robledo for the detailed review of the initial version of our manuscript.

Referee #2 (Remarks to the Author):

We thank Holzmann and colleagues for their thorough and constructive response to our earlier comments.

In our previous report, we raised concerns regarding the robustness of the phylogenetic framework used in the ancestral-state reconstruction of CT_{max} , particularly the reliance on COI barcodes for taxonomic assignment, uncertainty in tree topology and branch lengths, and the use of a family-level backbone based on a limited number of molecular markers. These issues are relevant because several insect families have since been shown to be paraphyletic, interfamily relationships have been revised by phylogenomic studies, and the taxon sampling in Ramford et al. (2014) does not fully overlap with the present dataset.

The authors have now undertaken a series of additional analyses to address these concerns:

1. Families now known to be non-monophyletic were excluded and the analyses repeated. Comparing CT_{max} estimates for six order-level nodes (Hymenoptera, Diptera, Lepidoptera, Coleoptera, Hemiptera, Orthoptera) and five family-level nodes (Braconidae, Chrysomelidae, Formicidae, Ichneumonidae, Muscidae) shows minimal change.
2. Morphology-based family assignments were generated for several orders and compared to COI-based assignments, yielding relatively high concordance (89% in East Africa; 91% in the

establishes a link between T_{max} and T_{min} trends in insects and addresses the phylogenetic constraints on heat resistance. Finally, the analysis of protein melting temperatures appears to be new to the ecological literature on climate change (although I am less qualified to assess its technical aspects and pitfalls).

The ecophysiological methods of T_{max} and T_{min} measurements are standards, as well as the approaches to the statistical evaluation of the data. The text is well written, the Figures are complex but clear and informative. The conclusions, based on large sets of data, are robust.

In summary, I feel that this is an important, timely, and highly informative and innovative work that addresses one of the most pressing issues in tropical insect ecology. I have a few minor questions and comments:

[1] The methods of sampling, especially in Peru, and the procedure used to arrive at 3,200 analysed species from 7,850 insect individuals need to be better described. Were the insects in Peru also sampled by sweep net? Were all individuals analysed for T_{max} and other parameters and then – after barcoding – consolidated to species level, so that some species represent mean values from several specimens? Were some obviously common species reduced to fewer individuals used for the measurements? This is not clear from the text. Also, the individual-to-species ratio of about 2.5 suggests very superficial (incomplete) sampling – understandable given the focus of the study, but it should be mentioned that the study is therefore probably limited to the more common species in the local insect communities.

[2] The climate scenarios used here are described in the literature, but it would still be useful to briefly characterise them in the online methods, and in particular to explain why the predicted temperature change differs so much between the two study regions.

[3] The paper briefly discusses microhabitat selectivity and other behavioural adaptations that could reduce the impact of the temperature measured in this paper. Are there any studies that attempt to quantify the potential of this high temperature avoidance for tropical forests? This could also explain the counter-intuitive finding that, according to the present analysis, some species are already struggling with heat stress while still surviving in the habitats studied. This paradox should be discussed in the paper.

Version 1:

Reviewer comments:

Relevée #1

(Remarks to the Author)

Dear editors,

I am very happy with the detailed response to the comments. The study is original and involves many insect species in a broad geographic range. Data and methodology are valid, and the authors ensured to explain the potential issues of phylogenetic analyses using DNA barcodes. The new version of the manuscript is clear and references are adequate. My recommendation is to accept in its current form.

--- Carlos Garcia-Robledo, Ph.D.

Associate Professor

Department of Ecology and Evolutionary Biology

University of Connecticut

Relevée #2

(Remarks to the Author)

We thank Holzmann and colleagues for their thorough and constructive response to our earlier comments.

In our previous report, we raised concerns regarding the robustness of the phylogenetic framework used in the ancestral-state reconstruction of CT_{max}, particularly the reliance on COI barcodes for taxonomic assignment, uncertainty in tree topology and branch lengths, and the use of a family-level backbone based on a limited number of molecular markers. These issues are relevant because several insect families have since been shown to be paraphyletic, interfamily relationships have been revised by phylogenomic studies, and the taxon sampling in Rainford et al. (2014) does not fully overlap with the present dataset.

The authors have now undertaken a series of additional analyses to address these concerns:

1. Families now known to be non-monophyletic were excluded and the analyses repeated. Comparing CT_{max} estimates for six order-level nodes (Hymenoptera, Diptera, Lepidoptera, Coleoptera, Hemiptera, Orthoptera) and five family-level nodes (Braconidae, Chrysomelidae, Formicidae, Ichneumonidae, Muscidae) shows minimal change.
2. Morphology-based family assignments were generated for several orders and compared to COI-based assignments, yielding relatively high concordance (69% in East Africa, 91% in the Neotropics). Re-running analyses using morphology-based assignments resulted in only ~2% average change in order-level CT_{max} estimates, with the elevation trend preserved.
3. Families present in the current dataset but not in Rainford et al. (2014) were previously inserted adjacent to sister families. Reanalysis excluding these manually placed families produced an average change of 0.04% in CT_{max} estimates.
4. The use of a maximum-likelihood tree with bootstrap support, instead of the earlier neighbor-joining tree, provides a more

Guralnick, N. E. Pierce, and D. J. Lohman. 2023. A global phylogeny of butterflies reveals their evolutionary history, ancestral hosts and biogeographic origins. *Nature Ecology & Evolution* 7(6):903-913. doi: 10.1038/s41559-023-02041-9

Kortmann, M., A. Chao, C.-H. Chiu, C. Heibl, O. Mitesser, J. Morinière, V. Bozicevic, T. Hothorn, J. Rothacher, and J. Englmeier. 2025. A shortcut to sample coverage standardization in metabarcoding data provides new insights into land-use effects on insect diversity. *Proceedings B* 292(2046):20242927.

Lambkin, C. L., B. J. Sinclair, T. Pape, G. W. Courtney, J. H. Skevington, R. Meier, D. K. Yeates, V. Blagoderov, and B. M. Wiegmann. 2013. The phylogenetic relationships among infraorders and superfamilies of Diptera based on morphological evidence. *Systematic Entomology* 38(1):164-179. doi: <https://doi.org/10.1111/j.1365-3113.2012.00652.x>

Mildrexler, D. J., M. Zhao, and S. W. Running. 2011. A global comparison between station air temperatures and MODIS land surface temperatures reveals the cooling role of forests. *Journal of Geophysical Research: Biogeosciences* 116(G3)doi: <https://doi.org/10.1029/2010JG001486>

Rainford, J. L., M. Hofreiter, D. B. Nicholson, and P. J. Mayhew. 2014. Phylogenetic Distribution of Extant Richness Suggests Metamorphosis Is a Key Innovation Driving Diversification in Insects. *PLoS ONE* 9(10):e109085. doi: 10.1371/journal.pone.0109085

Scheffers, B. R., T. A. Evans, S. E. Williams, and D. P. Edwards. 2014. Microhabitats in the tropics buffer temperature in a globally coherent manner. *Biology Letters* 10(12):20140819. doi: 10.1098/rsbl.2014.0819

Song, H., O. Béthoux, S. Shin, A. Donath, H. Letsch, S. Liu, D. D. McKenna, G. Meng, B. Misof, L. Podsiadlowski, X. Zhou, B. Wipfler, and S. Simon. 2020. Phylogenomic analysis sheds light on the evolutionary pathways towards acoustic communication in Orthoptera. *Nature Communications* 11(1):4939. doi: 10.1038/s41467-020-18739-4

Song, N., M.-M. Wang, W.-C. Huang, Z.-Y. Wu, R. Shao, and X.-M. Yin. 2024. Phylogeny and evolution of hemipteran insects based on expanded genomic and transcriptomic data. *BMC Biology* 22(1):190. doi: 10.1186/s12915-024-01991-1

Sunday, J., J. M. Bennett, P. Calosi, S. Clusella-Trullas, S. Gravel, A. L. Hargreaves, F. P. Leiva, W. C. E. P. Verberk, M. Á. Olalla-Tárraga, and I. Morales-Castilla. 2019. Thermal tolerance patterns across latitude and elevation. *Philosophical Transactions of the Royal Society B: Biological Sciences* 374(1778):20190036. doi: 10.1098/rstb.2019.0036

Wiens, J., R. Lapoint, and N. Whiteman. 2015. Herbivory increases diversification across insect clades. *Nat Commun* 6: 8370.

robust phylogenetic framework.
5. The authors introduced branch-length perturbations (10% jitter) across 50 replicate trees to evaluate variation in ancestral-state reconstruction. We would appreciate clarification on the choice of the 10% perturbation level and whether alternative magnitudes (e.g., 5%, 20–50%) were considered or tested.

6. Randomly removing 50% of taxa in 100 replicate analyses produced CTmax estimates congruent with the full dataset at both order- and family-level nodes.

Overall, we find that the additional analyses substantially strengthen the study and address the primary concerns we raised. The conclusions now appear more robust to taxonomic assignment differences, phylogenetic uncertainty, and sampling variation.

Xin Zhou and Guanliang Meng

Referee #3

(Remarks to the Author)

I co-reviewed this manuscript with one of the reviewers who provided the listed reports.

Referee #4

(Remarks to the Author)

The authors have addressed all my concerns where possible and the inclusion of a multi-model ensemble for climate change projections now delivers the required uncertainty assessment with respect to climate change too. I do not have further concerns regarding this study.

Referee #5

(Remarks to the Author)

I am satisfied with the response to my specific comments as well as with the revised manuscript as a whole.

Referee #6

(Remarks to the Author)

Holzmann et al. explore the evolutionary patterns of upper thermal tolerance (CTmax) across insects and consider the extent to which these traits may respond to climate warming. Their analyses indicate that CTmax variation is shaped more strongly by shared evolutionary history than by present-day thermal conditions. This pattern implies a considerable degree of phylogenetic conservatism, with closely related lineages exhibiting comparable upper thermal limits. The authors substantiate this inference through the detection of a pronounced phylogenetic signal in CTmax and by reconstructing ancestral thermal states.

My review was requested to concentrate specifically on the thermal stability prediction and its evaluation. From this standpoint, I identify a methodological limitation that, in my view, does not alter the main conclusions. The tool used by the authors, DeepSTABp, is well suited for predicting distributions of protein thermal stability based on structural properties. However, when applied across species, additional factors, such as differences in compatible solutes or other aspects of the cellular protein environment, may influence protein stability. DeepSTABp attempts to account for such effects by incorporating optimal growth temperature as an additional parameter.

In the present study, the authors use the same temperature for all predictions. Because their analysis focuses on differences in the lower quartile of the predicted stability distributions, the observed effects remain valid. Moreover, their comparative analyses are most likely robust to this assumption due to aggregation, which I find to be a thoughtful and effective analytical choice. While the absolute temperature values on the y-axis may be shifted in a more or less linear fashion, such a shift would not affect the qualitative pattern or the central findings of the study.

Therefore, I agree with the authors' interpretation and consider their conclusions to be well supported despite the noted methodological limitation.

References:

- Ashe-Jepson, E., S. Arizala Cobo, Y. Basset, A. J. Bladon, I. Kleckova, B. C. Laird-Hopkins, A. McFarlane, K. Sam, A. F. Savage, A. C. Zamora, E. C. Turner, and G. P. A. Lamarre. 2023. Tropical butterflies use thermal buffering and thermal tolerance as alternative strategies to cope with temperature increase. *Journal of Animal Ecology* 92(9):1759-1770. doi: <https://doi.org/10.1111/1365-2656.13970>
- Baena-Bejarano, N., C. Reina, D. E. Martínez-Revelo, C. A. Medina, E. Tovar, S. Uribe-Soto, J. C. Neita-Moreno, and M. A. Gonzalez. 2023. Taxonomic identification accuracy from BOLD and GenBank databases using over a thousand insect DNA barcodes from Colombia. *PLOS ONE* 18(4):e0277379. doi: [10.1371/journal.pone.0277379](https://doi.org/10.1371/journal.pone.0277379)
- Bennett, J. M., J. Sunday, P. Calosi, F. Villalobos, B. Martínez, R. Molina-Venegas, M. B. Araújo, A. C. Algar, S. Clusella-Trullas, B. A. Hawkins, S. A. Keith, I. Kühn, C. Rahbek, L. Rodríguez, A. Singer, I. Morales-Castilla, and M. A. Olalla-Tárraga. 2021. The evolution of critical thermal limits of life on Earth. *Nature Communications* 12(1):1198. doi: [10.1038/s41467-021-21263-8](https://doi.org/10.1038/s41467-021-21263-8)
- Blaimer, B. B., B. F. Santos, A. Cruaud, M. W. Gates, R. R. Kula, I. Mikó, J.-Y. Rasplus, D. R. Smith, E. J. Talamas, S. G. Brady, and M. L. Buffington. 2023. Key innovations and the diversification of Hymenoptera. *Nature Communications* 14(1):1212. doi: [10.1038/s41467-023-36868-4](https://doi.org/10.1038/s41467-023-36868-4)
- Bujan, J., and S. P. Yanoviak. 2022. Behavioral response to heat stress of twig-nesting canopy ants. *Oecologia* 198(4):947-955.
- Cai, C., E. Thielka, M. Giacomelli, J. F. Lawrence, A. Šlipiński, R. Kundrata, S. Yamamoto, M. K. Thayer, A. F. Newton, R. A. B. Leschen, M. L. Gimmel, L. Lü, M. S. Engel, P. Bouchard, D. Huang, D. Pisani, and P. C. J. Donoghue. 2022. Integrated phylogenomics and fossil data illuminate the evolution of beetles. *Royal Society Open Science* 9(3):211771. doi: [doi:10.1098/rsos.211771](https://doi.org/10.1098/rsos.211771)
- Jørgensen, L. B., H. Malte, M. Ørsted, N. A. Klahn, and J. Overgaard. 2021. A unifying model to estimate thermal tolerance limits in ectotherms across static, dynamic and fluctuating exposures to thermal stress. *Scientific Reports* 11(1)doi: [10.1038/s41598-021-92004-6](https://doi.org/10.1038/s41598-021-92004-6)
- Jørgensen, L. B., M. Ørsted, H. Malte, T. Wang, and J. Overgaard. 2022. Extreme escalation of heat failure rates in ectotherms with global warming. *Nature* 611(7934):93-98.
- Joshi, C. H., and J. J. Wiens. 2023. Does haploploidy help drive the evolution of insect eusociality? *Frontiers in Ecology and Evolution* 11:1118748.
- Kawahara, A. Y., C. Storer, A. P. S. Carvalho, D. M. Plotkin, F. L. Condamine, M. P. Braga, E. A. Ellis, R. A. St Laurent, X. Li, V. Barve, L. Cai, C. Earl, P. B. Frandsen, H. L. Owens, W. A. Valencia-Montoya, K. Aduse-Poku, E. F. A. Toussaint, K. M. Dexter, T. Doleck, A. Markee, R. Messcher, Y. L. Nguyen, J. A. T. Baddon, H. A. Benítez, M. F. Braby, P. A. C. Buenavente, W.-P. Chan, S. C. Collins, R. A. Rabideau Childers, E. Dankowicz, R. Eastwood, Z. F. Fric, R. J. Gott, J. P. W. Hall, W. Hallwachs, N. B. Hardy, R. L. H. Sipe, A. Heath, J. D. Hinolan, N. T. Homziak, Y.-F. Hsu, Y. Inayoshi, M. G. A. Itliong, D. H. Janzen, I. J. Kitching, K. Kunite, G. Lamas, M. J. Landis, E. A. Larsen, T. B. Larsen, J. V. Leong, V. Lukhtanov, C. A. Maier, J. I. Martínez, J. J. Martins, K. Maruyama, S. C. Maunsell, N. O. Peda, A. Monastyrskii, A. B. B. Morais, C. J. Müller, M. A. K. Naive, G. Nielsen, P. S. Padrón, D. Peggie, H. P. Romanowski, S. Sáfián, M. Saito, S. Schröder, V. Shirey, D. Solits, P. Solits, A. Sourakov, G. Talavera, R. Vila, P. Vlasanek, H. Wang, A. D. Warren, K. R. Willmott, M. Yago, W. Jetz, M. A. Jarzyna, J. W. Breinholt, M. Espeland, L. Ries, R. P.

physiological limit. We found one study that elucidated behavioral heat avoidance of canopy ants in tropical forests (Bujan and Yanoviak, 2022). We have added these thoughts and references also to the discussion on thermal sensitivity in the main manuscript and now write: “Tropical butterflies, for instance, have shown a thermal buffering ability up to 1 °C (Asherson et al., 2023), and twig-nesting ants abscond their nests under heat stress (Bujan and Yanoviak, 2022). We expect that avoidance of stressful temperatures by thermoregulatory behaviour is possible if, first, complex habitat structures exist that provide shade, which can buffer temperatures up to 4 °C (Scheffers et al., 2014), and, second, shade air temperatures remain below the injury causing threshold.”

In cases where reviewers are anonymous, credit should be given to 'Anonymous Referee' and the source. The images or other third party material in this Peer Review File are included in the article's Creative Commons license, unless indicated otherwise in a credit line to the material. If material is not included in the article's Creative Commons license and your intended use is not permitted by statutory regulation or exceeds the permitted use, you will need to obtain permission directly from the copyright holder.

To view a copy of this license, visit <https://creativecommons.org/licenses/by/4.0/>.

Response to Referees

Referee comments are shown in light grey, author responses in black

Referee #1 (Remarks to the Author):

A. Summary of the key results. In the manuscript “Limited thermal tolerance in tropical insects and its genomic signature”, the authors explore how thermal tolerance changes with elevation along two elevational gradients in Peru and Kenya. The study reports critical thermal maxima and minima for more than 2000 “species” identified using the DNA barcode COI.

Using COI sequences, the authors generated a phylogeny to test for phylogenetic signal in CTmax among clades. Experiments exposing insects to extreme temperatures (shock) and then to increasing temperatures aimed to determine plasticity in thermal tolerance among species. Using published databases, the study includes thermal responses of heat shock proteins in various insects. The study includes a projection of how future temperatures may generate thermal coma.

B. Originality and significance: The study includes an outstanding number of species along two elevational gradients. The main result, that CTmax decreases with elevation in tropical insect species was previously published. The phylogenetic analysis, heat shock protein analyses, and the extinction risk forecasting analysis are interesting.

C. Data & methods: The authors delimited species using the DNA barcode COI, using sequences with “more than 100 bp”. This is problematic. If many of the included sequences are short (i.e., close to 100 – 300 bp) it is possible that this will inflate the number of MOTUS (i.e., identify as another species individuals with sequences too short to match sequences close to ~600 bp).

KLJH: Thank you for reviewing our manuscript, we greatly appreciate your effort and your help in improving the quality of our study.

We see that our description was misleading and acknowledge that the information we previously had provided on the barcoding method must be described in more detail. The COI sequences had an average length of 557 bp with a minimum overlap of 303 bp. Ca. 75% of all sequences had an overlap of more than 500 bp, and half of the sequences overlapped with more than 600 bp. The 100 bp was just the BOLD default filter we had set. We have added methodological details now to the methods to make this clearer.

These results, as well as detailed information on genetic clustering, have been added now to the Supplementary Methods and are displayed in the Extended Data Table 2. We have clarified the methods regarding the collection of insects and now write: “Insects of all major orders (mainly Coleoptera, Diptera, Hymenoptera and Lepidoptera; additionally, Hemiptera and Orthoptera) were collected in the Neotropics (n = 4,690) in three seasons (September to December 2022, April to August 2023, and September to December 2023) with sweep nets in the understory of all study plots. In the Afrotropics, insects (n = 3,164) were collected in one season (March to June 2023) applying the same method.”

We now also acknowledged the limitations of our sampling in the main manuscript and write: “Insect collection was focused on taxonomic breadth covering six major orders rather than on individual species. Note, that our results represent the more common insect communities. In the Supplementary Methods, we provide extensive robustness analyses to assess the effects of incomplete sampling or modifications to the phylogeny.”

[2] The climate scenarios used here are described in the literature, but it would still be useful to briefly characterise them in the online methods, and in particular to explain why the predicted temperature change differs so much between the two study regions.

KLJH: We agree that more information on the climate change projections is useful and have added detailed information on the SSPs and different climate models to the Supplementary Methods; we referred to this detailed characterization in the main text. Here, we also provide the likely explanation of the carbon cycle feedback for the greater temperature anomalies in the Amazonian region. We also added a short overview of the scenarios in the online methods and write: “For future climate projections (2071–2100) under the three shared socio-economic pathways SSP1-2.6 (sustainable scenario), SSP3-7.0 (medium-high scenario), and SSP5-8.5 (high scenario) we used the GFDL-ESM4 climate model predictions.”

[3] The paper briefly discusses microhabitat selectivity and other behavioural adaptations that could reduce the impact of the temperature measured in this paper. Are there any studies that attempt to quantify the potential of this high temperature avoidance for tropical forests? This could also explain the counter-intuitive finding that, according to the present analysis, some species are already struggling with heat stress while still surviving in the habitats studied. This paradox should be discussed in the paper.

KLJH: Thank you for this valuable comment. We fully agree that behavioral avoidance of high temperatures is a very interesting and important topic to discuss in this context. There are some studies on the topic, but these are highly limited to only few species making generalizations difficult. For example, a study on tropical butterflies in Panama, showing the capability of thermal buffering up to 1 °C in Pieridae (Ashe-Jepson et al., 2023). Furthermore, studies illustrate that micro-habitats can buffer temperatures by ~4°C (Scheffers et al., 2014), allowing organisms to habituate environments that reach temperatures beyond their

Referee #5 (Remarks to the Author):

This is the most interesting, comprehensive and innovative manuscript I have seen in the recently emerging wave of studies on thermal tolerance, especially in the tropics. This research topic is a response to the increasing concern about climate change and, in particular, insect attrition in lowland tropical areas, which was raised as a highly controversial possibility almost 20 years ago by Colwell et al. (2008) (although the present paper does not cite this study in this context). This paper is the clearest indication yet that Colwell et al. may be right and that tropical lowland insects may be at serious risk.

The reason why this paper stands out from other similar studies is its comprehensive dataset, which includes more than 2,000 insect species along two long tropical gradients on two continents. In addition, the question of plasticity of T_{max} is partly answered by measuring its changes after heat shock, presumably following upregulation of heat shock proteins. It also establishes a link between T_{max} and T_{min} trends in insects and addresses the phylogenetic constraints on heat resistance. Finally, the analysis of protein melting temperatures appears to be new to the ecological literature on climate change (although I am less qualified to assess its technical aspects and pitfalls).

The ecophysiological methods of T_{max} and T_{min} measurements are standards, as well as the approaches to the statistical evaluation of the data. The text is well written, the Figures are complex but clear and informative. The conclusions, based on large sets of data, are robust.

In summary, I feel that this is an important, timely, and highly informative and innovative work that addresses one of the most pressing issues in tropical insect ecology. I have a few minor questions and comments:

KLJH: Thank you, we highly appreciate your acknowledgement of our study.

[1] The methods of sampling, especially in Peru, and the procedure used to arrive at 3,200 analysed species from 7,850 insect individuals need to be better described. Were the insects in Peru also sampled by sweep net? Were all individuals analysed for T_{max} and other parameters and then – after barcoding – consolidated to species level, so that some species represent mean values from several specimens? Were some obviously common species reduced to fewer individuals used for the measurements? This is not clear from the text. Also, the individual-to-species ratio of about 2.5 suggests very superficial (incomplete) sampling – understandable given the focus of the study, but it should be mentioned that the study is therefore probably limited to the more common species in the local insect communities.

KLJH: The insects in Peru were collected following the same method than in Kenya. Indeed, all collected individuals were tested for their thermal limits and after barcoding, values were clustered by genetic units, i.e. “species”. You pointed out correctly that our sampling focused on the breadth covering six major insect orders rather than species-specific analyses. In response to another reviewer, we have now run extensive robustness analyses, excluding single counts of species and also randomly removing 50% of the data with 100 repetitions.

The authors must be cautious with their interpretation of the phylogenetic analysis. Using DNA barcodes to reconstruct ancestral characters such as thermal tolerance is incorrect. Even with more extensive molecular information per species, it is problematic that the sampling of species and families amongst orders is limited to a few taxa along elevational gradients.

KLJH: We acknowledge that DNA barcode sequences (COI) offer limited phylogenetic resolution for deep nodes and are not suitable alone for reconstructing and interpreting ancestral traits. Therefore, we constructed supertrees by using a published backbone phylogeny with a resolution up to the insect family level and only used the DNA barcodes for reconstructing phylogenetic relationships for taxa below the family level. Nevertheless, we agree that phylogenetic trees of higher quality could be constructed with broader sequencing approaches (e.g. whole genome sequencing), but this was not feasible for the number of species analyzed in this study. We regard our phylogenetic tree as a phylogenetic hypothesis, that provides a broad and novel baseline for future studies.

We also agree that the reconstruction of ancestral traits would be less error-prone with more species and families completing the phylogenetic tree. However, we want to point out that the tree includes thermal trait and sequence data on much more species and families than any tree previously published. The tree includes data on 234 insect families, which represent more than one third of all described insect families. Despite its methodological imperfections it provides probably the best estimate of thermal traits for ancient nodes yet published. In the new version of the manuscript, we estimated the potential problem of incompleteness by randomly removing 50% of all species (OTUs) with 100 repetitions and checking if the reconstructed states of nodes at the order level and for the five largest families remain stable (see Table 1 below). Removing half of the data did on average lead to only 0.35% changes in the estimates of CT_{max} (details in Table 1) for nodes at the level of orders, e.g. Diptera remain the group with the lowest CT_{max} and Orthoptera with the highest. This data validation is now available in the new Supplementary Methods and the Extended Data Table 2. We have also addressed these limitations in more detail in response to reviewer #2.

Table 1: Ancestral state CT_{max} values for insect orders and major families for the full data and a subset with 50% of data randomly removed with 100 repetitions.

Group	Ancestral state CT_{max}	
	Full data	50% of samples removed (100 rep., mean \pm SD)
Hymenoptera	42.26314	42.37062 \pm 0.2628579
Diptera	40.26145	40.46429 \pm 0.3046916
Lepidoptera	41.31444	42.07025 \pm 0.7431772
Coleoptera	42.01704	42.22882 \pm 0.3885880
Hemiptera	41.81403	41.84437 \pm 0.2940025
Orthoptera	43.41890	43.42219 \pm 0.2903087
Braconidae	42.53200	42.71256 \pm 0.3804674
Chrysomelidae	44.07428	44.37189 \pm 0.7066789
Formicidae	44.90566	45.77228 \pm 0.7764050
Ichneumonidae	41.71519	41.90235 \pm 0.3060222
Muscidae	41.05457	40.99158 \pm 0.3206086

Fig. 5: Estimates of critical temperatures based on a multi-model ensemble. Percentage of surface (red) and air (blue) future temperatures that are critical (i.e. can cause heat coma in less than 8 hours) at the Neotropical (Peru, left box, a–i) and East African (Kenya, right box, j–t) elevation gradient based on multi-model climate change anomalies (includes the models GFDL-ESM4, IPSL-CM6A-LR, MPI-ESM1-2-HR, MRI-ESM2-0 and UKESM1-0-LL; see Supplementary Methods); for median CT_{max} values (upper row), more heat-sensitive insects (25% quantile of CT_{max} , middle row) and for the most heat-sensitive insects (10% quantile of CT_{max} , lower row). The three shared socio-economic pathways SSP1-2.6, SSP3-7.0, and SSP5-8.5 are shown for each geographic region. Critical temperatures were calculated along the whole gradient assuming the CT_{max} from lowland organisms (Methods).

Clarity and Context:

The manuscript is generally well written and well structured. The abstract, introduction, and conclusions clearly state the motivation and findings. However, the climate projection methods, particularly how future temperatures are generated, require greater detail and clarity.

KLH: Thank you. Addressed, see details in earlier response.

3) Clarify in the methods that the surface temperature data (from ECOSTRESS) reflect land surface (skin) temperatures and not air temperatures, and justify the assumption that these warm at the same rate as CHELSA air temperature anomalies.

KLH: We have clarified this. See details in our earlier response.

4) The authors rely on a single global climate model (GFDL-ESM4) from the CMIP6 archive, following the ISMIP3b protocol, to calculate future climate anomalies (Abio5). While this model is widely used and part of an established impact protocol, relying on a single GCM does not capture the full range of uncertainty inherent in future climate projections. Inter-model variability, particularly at regional and elevational scales, can significantly influence ecological forecasts. I recommend that the authors incorporate an ensemble of CMIP6 models, such as those available through the CHELSA multi-model ensemble.

KLH: Thank you for pointing this out. We cautiously chose the GFDL-ESM4 model in the first place for our main manuscript, since it is shown to be most suitable for forested areas and has strength on carbon cycle feedback which is particularly important for Amazonian lowland areas. Without considering this, an underestimation of future climate warming is very likely. We added now a sentence justifying the focus on this model in the main text and added an extra section in the Supplementary Methods presenting more details. We re-ran our climate change projections with all available CHELSA climate models and present the results in a new comprehensive figure (see below + Extended Data Fig. 10). Considering a multi-model ensemble instead of relying on a single model does not change the interpretation of our results. In the East African region, the projected future climate basically remains the same, in the Neotropics the anomaly under the other models is slightly lower but still leads to heat coma in a highly significant proportion of the insect community. We have added those alternative results in detail to the Supplement and refer to it in the main manuscript.

The supplementary material should include more details on the number of individuals per OTU. If the sampling was balanced, I would expect about 5 individuals per "species". However, it is very likely that OTUs represented by a single individual were common. The evenness of the sampling should be reported in the manuscript, as well as in the metadata and datasets associated with the paper. Analyses must include OTU as a factor to ensure that the results are not skewed by abundant species. If the authors included CT_{max} estimates based on a single individual, I think these data should be removed.

KLH: We have added detailed information on OTUs to the Methods and agree that this information clearly improves the methodological transparency of our manuscript. As expected from the sampling design from our study, covering a very broad range of insect taxa, we had on average 2 individuals per OTU. We also ran analysis of CT_{max} and CT_{min} along elevation with OTU as a factor, showing that OTU as random factor does not change the estimates or elevational pattern of thermal limits (Table 2).

Table 2: Comparison of GAM models of CT_{max} and CT_{min} along elevation. Model without random term: CT_{max}/CT_{min} ~ s(Elevation, k = 5) shown in black (upper term) and including a random term: CT_{max}/CT_{min} ~ s(Elevation, k = 5) + s(OTU, bs="re") shown in red (lower term).

	Area	Intercept	eef Elevation	eef OTU	P value
CT _{max}		41.426 41.321	3.162 3.243	556.166	< 0.001
		45.985 45.374	3.796 3.705	442.398	
CT _{min}		5.136 5.007	3.852 3.817	469.611	< 0.001
		5.854 5.891	3.417 1.01	218.67	

In a second set of analyses we removed OTUs represented by single individuals, resulting in a change of 0.72% in the estimates of CT_{max} at the level of orders (details in Tabel 3). Both approaches did not lead to significant changes in the reconstruction of ancestral trait states nor in elevational trends of thermal limits (Fig. 1). Excluding single OTUs leads to an even stronger phylogenetic signal within the CT_{max} data across the insect tree of life. From our perspective, however, an analysis including also OTUs with single individuals and without OTU as a random term is more complete, as the changes in CT_{max} of the insect assemblages with elevation is not only a consequence of changes in CT_{max} within species but mostly of species turnover. This is particularly true for hyperdiverse tropical rainforests, where species often have very small thermal niches and have tiny elevational ranges. Concerning both analyses we follow that the presented results are robust against the use of mixed effects models with species as a random term instead of linear models and against the exclusion of species only represented by single individuals. We added the results underscoring the robustness of our approach in the Extended Data Table 2 and Extended Data Fig. 3.

Table 3: Ancestral state CT_{max} values for insect orders and major families for the full data and a subset with single OTUs excluded.

Group	Ancestral state CT_{max}	
	Full data	Single OTUs excluded
Diptera	40.26145	40.74018
Hymenoptera	42.26314	42.93432
Lepidoptera	41.31444	42.74539
Coleoptera	42.01704	42.90529
Hemiptera	41.81403	41.12870
Orthoptera	43.41890	43.40575
Braconidae	42.53200	43.86825
Chrysomelidae	44.07428	45.53216
Formicidae	44.90566	45.81872
Ichneumonidae	41.71519	43.57961
Muscidae	41.05457	41.63238

➤ Pagels lambda as a proxy for the phylogenetic autocorrelation in the data

Full data: Pagel's $\lambda = 0.76$ ($p < 0.001$), Blomberg's $K = 0.33$ ($p < 0.001$); 10,000 randomizations)

Single OTUs excluded: Pagel's $\lambda = 0.96$ ($p < 0.001$), Blomberg's $K = 0.60$ ($p < 0.001$); 10,000 randomizations)

➤ theta parameter estimating the evolutionary optimum in CT_{max}

Full data: 42.20 °C. Single OTUs excluded: 42.37 °C

clearly addressed. Treatment of variance, extremes, and potential non-linear effects in warming scenarios could be improved.

KLH: We thank the reviewer for the feedback on our statistical approach. In the comment above we elaborate the microclimate and show that our field-measured temperature correlates strongly with the CHELSA data; hence, we believe it to serve as a robust baseline for anomaly calculations.

Conclusions: Robustness and Validity:

The overall conclusions are well supported by the experimental physiology and elevational comparison. However, the credibility of the future exposure analyses depends on addressing the climate projection concerns noted above. Until those are resolved, the broader ecological conclusions about future vulnerability remain tentative.

KLH: We have addressed concerns regarding the use of field-measured climate data in combination with CHELSA data in the earlier comment. We have now also added climate projections using an ensemble of all available climate models, showing that the overall conclusions are not affected even when the projected anomalies vary. More details are provided in our response to the fourth suggested improvement.

Suggested Improvements:

1) To better align microclimate data with the CHELSA climatology baseline, I recommend estimating the difference between open-air macroclimate (e.g., CHELSA or local weather station data) and below-canopy air temperature during the observed year. This offset can then be applied to both historical and future CHELSA values to generate baseline-aligned below-canopy projections.

KLH: We thank the reviewer for the constructive suggestion to improve our temperature projections. Unfortunately, suitable local weather data are not available for our remote field sites, and CHELSA does not provide gridded temperature data for 2022–2023. Higher temporal resolution datasets also exist only at very coarse spatial scales, which are unsuitable for steep elevational gradients. To increase transparency, we now show in the Supplementary Methods that our field-measured microclimate data strongly correlate with CHELSA long-term climate values, supporting their robustness as a baseline for future climate calculations. We have described this in detail in the earlier response.

2) Given the availability of measured daily temperature data, the authors could implement a quantile mapping approach rather than a simple additive delta-change. This would account for potential non-linearity in the relationship between air temperature and below-canopy microclimate, particularly at extremes.

KLH: We thank the reviewer for this suggestion, but due to the lack of current and future temperature data with both a high spatial and temporal resolution for the study region, quantile mapping is not possible. We have addressed this in the earlier comment on microclimate calculations.

KLH: We thank the reviewer for the thoughtful and constructive suggestion regarding the improvement of our below-canopy temperature projections. Unfortunately, however, our field sites are located in remote tropical forest areas without access to consistent local weather station data. In addition, CHELSA does not currently provide gridded air temperature data for the years 2022–2023, which prevents direct comparison between our recent canopy-level measurements and macroclimate air temperature for the same period. We also checked the availability of data with a higher temporal resolution in order to conduct quantile mapping but all data sets with a higher temporal resolution are only available with a very low spatial resolution of approximately 25 km; for a steep elevation gradient this resolution is not usable as this corresponds e.g. in our study region in Peru to a 2.5 km gradient in elevation (with approx. 1.5°C difference in mean annual temperature). This also would mean that a majority of our study plots would fall into one single cell of the grid. Consequently, all available data sets are either not in a suitable resolution for elevational gradients, or do not cover the time period of our sampling.

However, to improve transparency on this we now show in the Supplementary Methods how the microclimate data measured with loggers on study sites relates to the long-term BIOCLIM+ climate data from CHELSA: The bio1 (mean annual temperature) from CHELSA (blue) and the mean annual temperature calculated from field-measured microclimate from the TMS loggers (red) have a very strong correlation of $r_{\text{Pearson}} = 0.994$. The bio5 (mean maximum temperature of the warmest month) values of CHELSA BIOCLIM+ and the same measure calculated from our data loggers have a correlation of $r_{\text{Pearson}} = 0.962$ (see Fig. 4 below, and Extended Data Fig. 9), which is why we believe that our field measured microclimate provides a robust baseline for future climate calculations. Furthermore, our climatic data covers the tropical “dry” season twice reducing the risk of capturing an extreme/unusual year.

Fig. 4: Air temperatures (a) bio1 and (b) bio5 from CHELSA (blue) and the field-measured microclimate from the TMS loggers (red) showed a very strong overlap.

Appropriate Use of Statistics and Treatment of Uncertainties: Statistical treatment of thermal physiology and model fitting appears appropriate. However, uncertainty in future temperature projections — particularly regarding the assumption that a single year of microclimate data can serve as a baseline for delta-change — should be more

Fig. 1: Elevational trends of (a) full data CT_{max} and (b) without single OTUs, and the same for (c) full data CT_{min} and (d) without single OTUs. Black = Afrotropics, grey = Neotropics.

The “protein stability prediction” analysis should be part of another manuscript. Data for these analyses were obtained online, not from the sampled insects. The connection between the arguments of thermal tolerance of different proteins and the study performed in the field is not clear.

KLH: We can remove this part from our manuscript if the editor or reviewer insists on it, but we believe that the protein stability predictions are a novel and exciting part of our study that add an important mechanistic explanation for the pattern we have observed empirically. We assume that this connection was not made very clear in previous version of the manuscript. To make the connection between thermal protein stability and thermal tolerance at the organism level clearer, we restructured the text on the thermal stability of proteins.

We now write: “Predicted T_m across insect orders and families were highly predictive of the observed CT_{max} values (Fig. 2). This suggests that a part of the phylogenetic variation in organism-level CT_{max} is due to fundamental differences in protein architecture across insect orders, which may have been optimized to different temperature levels in the early evolution of insects. Trends between CT_{max} and T_m were consistently found for analyses including all proteins, the 25% proteins with the lowest T_m per species and for analyses restricted to the 43 proteins shared in the randomly selected set of proteins across all orders (Extended Data Fig. 6).”

Using CT_{max} to model organismal responses to changing temperatures is problematic. CT_{max} is not fitness, and populations will decline at temperatures lower than those at which insects collapse in the laboratory. Using such extreme temperatures (e.g., heat coma temperatures) to project responses to warming may be misleading, and underestimates the effect of temperature on insect extinctions.

KLH: We fully agree that CT_{max} is only a proxy for population fitness, and that it captures just one component of a complex set of physiological and ecological factors. Our approach does not attempt to represent all mechanisms underlying thermal vulnerability but instead

focuses on quantifying a key physiological limit that can be measured with a standardized and comparable method. To improve the applicability of our results, we used a model that translates CT_{max} from short, dynamic ramping rates into estimates of heat coma time under longer, constant heat exposure. This better reflects the relationship between acute and chronic heat stress, bridging the gap between experimental assays and real-world conditions. While we acknowledge that sublethal damage may begin before the onset of coma, it is equally important to consider that insects in natural environments may use behavioral strategies, such as seeking shade or burrowing, to mitigate heat exposure. We therefore view our approach as one useful tool within a broader framework to assess climate-related extinction risk in insects. Our aim is to provide a scientifically grounded estimate of vulnerability that is both robust and transparent about its limitations. In the discussion we now point out the complexity of the response of insects to warming and the limitations of our study in this respect.

We have added: "However, please note that the response of insects to climate warming is highly complex and experimental assays can only provide a proxy to predict heat tolerance as populations may decline at temperatures even lower than those causing heat coma due to sublethal heat injuries (Jørgensen et al., 2021), and, because extreme heat events can lead to an escalated failure of biological processes even before CT_{max} of an organism is reached (Jørgensen et al., 2022).

Some minor comments

P4L61. The goal of the comparison with arctic species is not clear.

KLJH: We have clarified the sentence. We now write: "Thermal tolerance decreased from low to high elevation and was higher along the East African than along the Neotropical elevation gradient (Fig.1b). This suggests stronger adaptations of climatic niches in tropical insects compared to species with a Holarctic distribution, whose upper thermal limits did not vary with elevation in a global study (Sunday et al., 2019)."

P4L62. The mechanistic explanation for this statement is not clear.

KLJH: We have rewritten this statement and now write: "Long-term impacts of a less forested vegetation and higher exposition to heat in large parts of East Africa over the last million years have potentially shaped the greater heat tolerance of insect communities."

References. I noticed that some citations do not support the statements included in the manuscript. For example, use books and literature reviews to support very particular statements Please use primary literature (e.g. refs 13, 17). The text identifies studies as "meta-analyses when" they are not (e.g. refs 7,8,9).

KLJH: Thank you for your comment on the references, we have carefully double checked them. Originally reference 13, a book (Angilletta Jr, M.J. and M.J. Angilletta, Thermal adaptation: a theoretical and empirical synthesis. 2009: Oxford University Press.), was replaced with primary, empirical literature. Where we cited the synthesis before (ref. 17,

Detailed comments and responses

The authors also apply a delta-change approach using derived air temperature anomalies (bio5) to project future environmental temperatures. While this method is potentially appropriate for below-canopy air temperatures, the same anomalies are also applied to land surface temperatures derived from ECOSTRESS. This assumes that surface temperatures, representing radiative skin temperatures of the land or canopy, will increase at the same rate as air temperature, which may not hold true in forested environments where evapotranspiration, canopy structure, and radiation balance can modulate surface warming differently. I recommend the authors explicitly acknowledge this assumption and its limitations, particularly given the distinct physical processes governing air vs. surface temperature responses to climate change. Clarifying this in the methods and discussion would improve transparency and help readers interpret the results more cautiously.

KLJH: We appreciate the reviewer's thoughtful comment on the application of air temperature anomalies to ECOSTRESS-derived surface temperatures and agree that this must be clarified better in the manuscript. Indeed, in a global data set (Mildrexler et al., 2011), the difference between maximum land surface temperatures (LST) and maximum air temperatures (AT) increases with increasing temperatures (with $LST > AT$). We can expect that assuming an increase of AT by e.g. 2°C the increase in LST will be even higher than 2°C. The difference strongly depends on habitat type with forested vegetation showing less increases than open areas. Thus, our approach provides rather conservative estimates of future surface temperatures. We have now added clarifications in the methods ("We acknowledge that surface temperatures may not warm at the same rate as air temperatures. The difference between surface and air temperatures is expected to increase with increasing temperatures, particularly in open areas (Mildrexler et al., 2011). Hence, our method provides conservative estimates of future surface temperatures.") and discussion of the main manuscript ("These values are conservative estimates, assuming that future surface temperatures warm at similar rates than air temperatures (Methods).")

To improve the accuracy of future below-canopy air temperature projections, I suggest an alternative to the current delta-change method. Specifically, the authors could calculate the difference between open-air macroclimate temperature (e.g., CHELSA bio5 for 2022–2023 or local weather station data) and the observed below-canopy air temperatures during the same period. This would yield an empirical distribution or average offset that quantifies local microclimate buffering under current conditions. This offset could then be applied to CHELSA's historical (1981–2010) and future (2071–2100) bio5 values, thereby generating below-canopy temperature estimates that are both baseline-aligned and regionally adjusted for canopy effects. This would reduce the risk of bias due to interannual anomalies in the observed data and better reflect how microclimates are expected to warm relative to regional climate. Since the authors also have daily measurements of climate data, they could apply a quantile mapping rather than a simple delta change, accounting for the fact that air-temperature and below canopy temperatures might not always be highly correlated at temperature extremes, or nonlinear.

warming trends (e.g., El Niño). As a result, the projected future temperatures may be biased high or low depending on whether the sampling year was anomalously warm or cool.

KLH: We thank the reviewer for the suggestion. Unfortunately, neither local weather data nor CHELSA provide suitable coverage for our sites and sampling years, and higher-resolution datasets are too coarse for steep elevational gradients. To address this critic, we show in the Extended Data Fig. 9 and Supplementary Methods that our microclimate data measuring on all plots during the study time indeed strongly corresponds to CHELSAs bio1 and bio5 variables measured between 1981–2010, supporting their reliability as a baseline. We respond to this in detail below.

Additionally, while the delta-change method may be appropriate for below-canopy air temperatures, the same anomalies are also applied to land surface temperatures derived from ECOSTRESS. This assumes that surface temperatures — representing radiative skin temperatures — warm at the same rate as air temperatures, which may not be valid in forested environments due to differential radiation balance, evapotranspiration, and canopy feedbacks. These assumptions should be explicitly acknowledged and discussed.

KLH: We thank the reviewer for this helpful comment and agree that clarification was needed. Since surface temperatures generally increase more than air temperatures with warming, especially in open habitats, our approach is likely to provide slightly more conservative estimates of future surface temperatures. We have now clarified this in the methods and discussion. We respond to this in more detail below in the detailed comment section.

Main point: Thermal sensitivity under climate change
The authors apply long-term mean climate anomalies (Δ bio5 from CHELSA, based on 1981–2010 and 2071–2100 climatologies) directly to high-resolution microclimate data collected over a single year (2022–2023). This assumes that the observed year is representative of the historical climatological baseline, which is unlikely given recent global warming trends and interannual variability (e.g., El Niño). As a result, the projected future temperatures may be biased high or low depending on whether the sampling year was anomalously warm or cool. A more robust approach would be to baseline-correct the observed microclimate data to the 1981–2010 mean before applying the climate anomaly. While the authors acknowledge limited long-term microclimate data availability, explicitly addressing this assumption and its potential effects on future temperature estimates would strengthen the methodological transparency of the study.

KLH: As suggested, we have now explicitly addressed this assumption in the main manuscript and included details showing the robustness of the microclimate data in the Supplementary Methods. We respond to this in detail below.

Horowitz, M., Heat acclimation: phenotypic plasticity and cues to the underlying molecular mechanisms. *Journal of Thermal Biology*, 2001. 26(4): p. 357–363. , we have now cited the specific studies instead. We have changed the wording to correct that we have cited not only meta-analyses, thank you for pointing this out.

Referee #2 (Remarks to the Author):

Holzmann and colleagues seek to understand how the thermal maxima (CT_{max}) of insects has evolved across the insect tree of life and its evolutionary potential under climate changes. They found that the evolutionary history of insects rather than local temperature conditions have a stronger influence on the evolution of CT_{max} . In other words, the evolution of CT_{max} is constrained by the phylogenetic relatedness and closely related organisms show similar CT_{max} values. The authors argue that this is supported by the strong phylogenetic signal in CT_{max} and ancestral state reconstruction for CT_{max} .

Similar findings have been reported previously (e.g., <https://doi.org/10.1111/j.1365-2435.2012.02036.x> and <https://doi.org/10.1038/s41467-022-32953-2>), but were supported by limited data size or studies not-specifically focusing on insects (e.g., <https://doi.org/10.1038/s41467-021-21263-8>). In this study, Holzmann and colleagues addressed the problem by focusing on six insect orders.

I was asked to focus my evaluation on phylogenetic reconstruction. In this regard, the manuscript does show some caveats in methodology. The major concern here is the lack of robustness in the tree topology and relevant evolutionary time. The authors will need to either improve their phylogenetic reconstruction or demonstrate why the current methodology won't affect their conclusions. I will elaborate this in more details:

In a nutshell, the authors adopted a family-level tree from a published work and added COI NJ trees to each of the pre-defined family nodes. I have a few major concerns about this methodology.

KLH: We thank referee #2 for the critical comments about the construction and use of the phylogenetic tree. Referee #2 correctly points out some shortcomings of the rather simple methods used by us to construct a phylogenetic tree and to infer ancient states of CT_{max} . Below we show in a detailed way how we dealt with this critic. Principally, we conducted multiple additional tree constructions, by using subsets of the data, alternating branch lengths, and using different methods for tree constructions. In all these different analyses we verified (i) the relationship between CT_{max} and elevation; (ii) the reconstruction of ancient states in CT_{max} at major nodes; (iii) Pagels lambda as a proxy for the phylogenetic autocorrelation in the data; (iv) the theta parameter estimating the evolutionary optimum in CT_{max} . We added a comprehensive table in the Extended Data and detailed information in Supplementary

Methods showing the results of these analyses. In a nutshell, these analyses show that our results are robust against variation in tree construction, completeness of taxonomic sampling and family-level assignment.

1. The adopted “family root tree” is far from being robust. Although the Rainford et al. (2014) tree had an impressive 82.1% family converge for extant insects, this tree was built on just a few gene markers, which has been shown less effective by recent phylogenomics studies when compared with genome-level markers, especially when the focus is placed at deep-level relationships. Perhaps an even larger concern is that the Rainford et al. phylogeny was based on a chimeric dataset, i.e., the representative data matrix for a given family was “assembled from multiple individuals and species” to improve gene coverage among taxa. An in-depth discussion on the 2014 tree is clearly out of scope here. But the reason for comfortably adopting this tree for the present study needs to be justified. In fact, family-level relationships of many insect orders have been re-examined heavily in the past 10 years using large-scale genomic data, for example, Coleoptera (<https://doi.org/10.1098/rsos.211771>), Orthoptera (<https://doi.org/10.1038/s41467-020-18739-4>), Lepidoptera (<https://doi.org/10.1038/s41559-023-02041-9>), Hymenoptera (<https://doi.org/10.1038/s41467-023-36868-4>). The authors should assess how the family-level insect phylogeny from Rainford et al. (2014) differs from these latest studies and the potential influence on their conclusion.

KLH: We thank the reviewer for raising this important point. We fully acknowledge that phylogenetic hypotheses continue to improve, particularly through recent genome-scale studies that have refined relationships within individual insect orders such as Coleoptera, Orthoptera, Lepidoptera, and Hymenoptera. These studies provide valuable insights at the order level, but unfortunately, they do not offer a unified framework that spans all insect orders and families simultaneously, which is essential for our study. In contrast, the phylogeny of Rainford et al. (2014) remains the most comprehensive and robust family-level backbone currently available across the entire insect tree of life. While not without limitations, it provides a good combination of breadth and resolution and has already proven to be a reliable scaffold in multiple large-scale comparative analyses (Wiens et al., 2015; Joshi and Wiens, 2023). For this reason, we consider it a strong and justified choice for our work. However, we also see that it has, in comparison to more recent phylogenomic trees on certain orders, deficits concerning topology, in some orders like Hymenoptera, and exact branching lengths. In the new version of the manuscript, we justify the use of the Rainford et al. (2014) tree and quantify potential problems concerning problems of tree topology and branch length. For the revised manuscript, we have now added Supplementary Methods and a comprehensive Extended Data table considering various scenarios of incomplete or uncertain phylogenies, demonstrating that the major results and conclusion are not affected by changes on the family-level. Based on the recent studies suggested, we quantified how many families that appear monophyletic in the Rainford tree turned out to be polyphyletic in the named phylogenomic studies. Moreover, we ran additional models in which we excluded families that turned out to be paraphyletic and added the results to the Extended Data table, demonstrating stable conclusions. We also emphasize that our framework is not bound to Rainford et al. (2014); it is designed to be flexible and can easily accommodate an updated backbone as soon as a more

9

Referee #3 (Remarks to the Author):

I co-reviewed this manuscript with one of the reviewers who provided the listed reports.

KLH: Thank you for co-reviewing our manuscript and your help in improving our work.

Referee #4 (Remarks to the Author):

This study presents a compelling integration of experimental thermal tolerance data with elevation-based ecological fieldwork and climate projections, offering a novel framework to evaluate tropical insect vulnerability to future warming. The authors combine empirical measurements of critical thermal maxima with modeled exposure times under current and projected microclimatic conditions. I would suggest that this study is published after the points with respect to the climate projections I raise below are appropriately dealt with.

Originality and Significance:

The manuscript brings together physiological data, environmental microclimate modeling, and biodiversity-relevant climate scenarios in a way that is innovative and timely. To my knowledge, few studies integrate insect thermal physiology with high-resolution, elevation-based climate exposure modeling at this scale. The study addresses a pressing and underexplored question, how tropical insect communities may be affected by climate change across elevational gradients, and does so with a creative and potentially impactful framework in my opinion.

KLH: We sincerely thank you for the interest and acknowledgement of our study, and for your valuable help in improving our study by suggesting constructive additions concerning the climate change projections, which we have included in the revised manuscript. Our revisions include now a comprehensive Supplementary Information file with the chapter “*Estimates of future climate*”, where we explain the calculation of our baseline climate data, further explanations and justification for the choice of climate models. Furthermore, we have added robustness analyses based on a multi-model ensemble considering all available climate models from CHELSA and present these analyses visually in the new Extended Data Fig. 10. We show the results and respond to this in detail below under the detailed comments.

Data & Methodology: Validity and Quality:

While the experimental and field components of the study are well executed and clearly presented, the climate projection component requires revision. The authors apply long-term mean climate anomalies (Abio5 from CHELSA, based on 1981–2010 and 2071–2100 climatologies) directly to high-resolution microclimate data collected over a single year (2022–2023). This assumes that the observed year is representative of the historical climatological baseline, which is unlikely given interannual climate variability and recent

KLH: We thank the reviewer for highlighting the choice of evolutionary models. We have decided to compare Ornstein-Uhlenbeck (OU) and Brownian motion (BM) since they are biologically most meaningful and comparable to literature (Bennett et al., 2021). We deliberately focused on two contrasting evolutionary models — Brownian motion and an Ornstein-Uhlenbeck process — because they represent two mechanistically opposed hypotheses about trait evolution. Brownian motion describes unconstrained, diffusion-like evolution without upper limits, whereas the OU process captures evolution toward an optimum, implying a bounded trait space. This binary contrast allows a direct test of whether trait evolution is best described as unbounded drift or constrained adaptation. Other model formulations included in the fitContinuous function (e.g. EB, delta, rate-trend, white) modify the rate or phylogenetic structure of trait change, but they do not explicitly represent the presence or absence of a physiological or evolutionary boundary, which was the focus of our hypotheses. Therefore, we here stuck to the test of the two contrasting hypotheses, i.e. a BM versus OU model of evolution of CT_{max} , and we added an explanation to our methods. We write: “We compare the OU and BM model since they represent two mechanistically contrasting hypotheses about trait evolution and are comparable to literature (Bennett et al., 2021). Under a Brownian motion model, traits are assumed to have evolved under random evolutionary drift, under an Ornstein-Uhlenbeck model with stabilizing selection towards an optimum.”

Minor issues:

Line 102: When assessing the amount of phylogenetic signal in the trait CT_{max} , the authors tested only Pagel's λ but not Blomberg et al.'s (2003) K , without providing corresponding p -values and simulation numbers (should be $\geq 10,000$). Please clarify.

KLH: Thanks for pointing this out. We have now refined our methods to calculate the phylogenetic signal and report now both Pagel's lambda and Blomberg's K including p values and 10,000 randomizations in the main manuscript.

Line 403: The application of a cutoff $\geq 80\%$ for family assignment needs to be justified or backed up by a reference.

KLH: We have added a recent reference showing that in BOLD an 80% family match is valid across all major insect orders (Baena-Bejarano et al., 2023).

Line 406: Reference 48 should be updated. The manuscript has been published.

KLH: Done, thank you for noticing.

comprehensive family-level phylogeny of all insects becomes available. In this way, our conclusions rest on the best currently available synthesis while remaining fully adaptable to future advances.

2. The family assignments for newly collected insects need to be carefully validated. Here, the authors focus on six insect orders, Coleoptera, Diptera, Hymenoptera, Lepidoptera, Hemiptera and Orthoptera. When possible, the authors assigned individual insects (collected from Kenya and Peru) to order- or family-level taxonomy based on morphology, before the specimens were sequenced for COI barcodes. Afterwards, the barcodes were clustered into OTUs, where the COI sequences were compared against the BOLD system for a family-level assignment. The authors employed an arbitrary BOLD similarity cutoff ($\geq 80\%$) for this purpose (Line 403). Unfortunately, family assignment based on COI barcodes is known to be less effective when compared to those of lower taxonomic hierarchies (genus, species), especially when closely related species was not present in the reference library. In this case, it is not uncommon that OTUs can be assigned to wrong families. I would argue that mistakes in family placement will have a significant impact on the interpretation of trait evolution, which is even worse than using a questionable phylogeny. As a general practice, the family level identification needs to be supported by morphology. If a consensus is indeed reached between morphology and barcode ID, this needs to be clarified and a summary of the results showing the congruence/discrepancy in this process should be presented.

KLH: We thank the reviewer for raising this important point regarding the validation of family-level assignments. As noted, many of our primary analyses (e.g. tests on elevational patterns in thermal tolerance, tests on plasticity, correlation between protein thermal stability and organism thermal tolerance) were conducted at the order level, which was morphologically confirmed for all specimens and therefore remain robust to potential errors in deeper taxonomic placement. However, to address this concern, for many specimens in several orders we have now additionally identified families morphologically and evaluated the congruence between morphological family assignments (for which we collected family level information morphologically) and those obtained through COI barcode matching. We found a high level of agreement: 89% (1477 matches) in East Africa and 91% (1174 matches) in the Neotropics. As a small proportion of discrepancies remained, we conducted additional analyses in which we included only the data with morphological family level assignments. They show that overall, the trend of CT_{max} with elevation remains stable, with an average 2% change in the estimates of CT_{max} at the level of orders (details in Table 4, Fig. 2 + Extended Data Fig. 3), the phylogenetic signal is still significant while being slightly stronger, and the evolutionary optimum in CT_{max} is very similar to estimates based on the full data set (details below + Extended Data Table 2). We have clarified this point in the revised manuscript and now explicitly summarize the congruence and discrepancy between morphological and barcode-based family IDs, as suggested by the reviewer. All results of the robustness analysis are now presented in the Supplementary Methods and Extended Data. After discussing problems of morphological and DNA-barcode based identification with taxonomic experts, we believe that the problem of including potentially wrongly assigned families by DNA barcoding is rather low and similarly high than the problem that would occur by a

morphology-based assignment. Taxonomic experts were clear that for many of the more scarce insect families that occur in a tropical Amazonian rainforests one would need a large number of specialized taxonomists that are hardly available; more important, experts stated that they usually see a very high and trustworthy agreement between morphology and DNA-barcode based family assignment; third, from their experience, morphological assignments, even by experts, also repeatedly fail to be correct when it comes to the identification of insects in hyperdiverse tropical environments with similar failure rates among the two methods. We therefore decided in the main text for the incorporation of all insect samples.

In the Extended Data (here also briefly shown), we added analyses from a data subset in which only insect samples were included in which morphological and DNA barcode identification matched (i-iv).

(i) the relationship between CT_{max} and elevation

Fig. 2: Elevational trends of (a) full data CT_{max} and (b) with only families where morphological and barcode ID match. The same for (c) full data CT_{min} and (d) matching samples only. Black = Afrotropics, grey = Neotropics.

(ii) the reconstruction of ancient states in CT_{max} at major nodes

Table 4: Ancestral state CT_{max} values for insect orders and major families for the full data and a subset excluding families without matching morphological and genetic identification.

	Ancestral state CT_{max}	
Group	Full data	Matching families
Diptera	40.26145	41.63869
Hymenoptera	42.26314	43.69528
Lepidoptera	41.31444	43.66797
Coleoptera	42.01704	44.05101
Hemiptera	41.81403	42.73412
Orthoptera	43.41890	43.50576
Braconidae	42.53200	43.86252
Chrysomelidae	44.07428	45.06098
Formicidae	44.90566	46.59996
Ichneumonidae	41.71519	42.96421
Muscidae	41.05457	40.49712

not affected; we included the test concerning variation in estimated branch length in the Supplementary Methods and Extended Data Table 2.

Table 7: Ancestral state CT_{max} values for insect orders and major families for the full data and data including random 10% branch length variation with 50 repetitions.

	Ancestral state CT_{max}	
Group	Full data	Jitter (50 rep., mean \pm SD)
Hymenoptera	42.26314	42.21016 \pm 0.02382502
Diptera	40.26145	40.09343 \pm 0.09389580
Lepidoptera	41.31444	41.19822 \pm 0.15115756
Coleoptera	42.01704	41.91576 \pm 0.15658274
Hemiptera	41.81403	41.68420 \pm 0.02978618
Orthoptera	43.41890	43.73603 \pm 0.25357627
Braconidae	42.53200	42.54435 \pm 0.08446263
Chrysomelidae	44.07428	43.98777 \pm 0.46062094
Formicidae	44.90566	44.92149 \pm 0.09126716
Ichneumonidae	41.71519	41.69933 \pm 0.02636138
Muscidae	41.05457	41.05068 \pm 0.02199536

6. Incomplete taxon sampling can also affect the outcomes of comparative analyses, such as ancestral state reconstruction. One approach to address this is to sequentially introduce hypothetical missing taxa into the phylogenetic tree and re-run the analyses on these expanded phylogenies to evaluate how ancestral state reconstruction is impacted (e.g., <https://doi.org/10.1073/pnas.0811421106>). Alternatively, researchers can randomly remove taxa from various clades—particularly those that are well-sampled—and examine how this reduction in taxon sampling influences the inference of ancestral states (e.g., <https://doi.org/10.1111/ibi.12211>).

KLH: We thank the reviewer for this important comment on taxon sampling. We have addressed this issue also in response to reviewer #1. Basically, we validated the robustness of our results by randomly removing 50% of the data in 100 repetitions, showing that the ancestral states of CT_{max} and its elevational trend remain stable, and all results lead to the same conclusion. The results are presented in a detailed table in the Extended Data Table 2 and described in the Supplementary Methods.

7. Finally, the authors tried to understand the evolution of the trait CT_{max} using the OU and BM models. More specifically, the authors tried to test if this trait evolves randomly or is under stabilizing selection towards an optimum. However, there are many trait evolution models (e.g., BM, OU, EB, lambda, rate, trend, delta and white) available and the rationale behind the model choice needs to be justified. This can be done with the fitContinuous() function in the package geiger. And ancestral state reconstruction should be carried out with the best-fit model.

Full data: Pagel's $\lambda = 0.76$ ($p < 0.001$), Blomberg's $K = 0.33$ ($p < 0.001$); 10,000 randomizations), theta parameter: 42.20 °C

Non-monophyletic families excluded: Pagel's $\lambda = 0.77$ ($p < 0.001$), Blomberg's $K = 0.33$ ($p < 0.001$), 10,000 randomizations), theta parameter: 42.19 °C

Another relevant issue is the use of the NJ method in sub-tree construction. This is problematic at least from a phylogenetics perspective. A character-based method, for example a maximum likelihood tree should instead be applied here. In addition, any inferred tree should be provided with support values to show how reliable the tree is (e.g., bootstrap values for the nodes).

KLH: We are grateful to the reviewer for this thoughtful and constructive comment. The subtrees representing families (n=242) are now based on the maximum likelihood method, and we provide support values for all multi-branch subtrees in the data repository (folder "aBayes"). Specifically, we report the aBayes values, which indicate the statistical support for each branch and thus address the concern regarding reliability of the inferred relationships. We would also like to refer to the bootstrap values of the backbone tree provided in Rainford et al. (2014). The methods were updated in the new version of the main manuscript.

5. A relevant question is the estimation on branch length of the new tree that integrated the "root" topology from Rainford et al (2014) and "leaves" from newly collected insects. In a phylogenetic comparative analysis including ancestral state reconstruction, a time tree is usually used as the input. The robustness of the topology and branch length of the input tree is important as many trait evolution models (e.g., the Brownian Motion model) attempt to model trait state (or value) changes over time (or proportional to time). An appropriate comparative method should recognize uncertainties associated with both topology and branch lengths, and subsequently incorporate these considerations into the analysis. (page41, <https://lukeharmon.github.io/pem/pdf/phylogeneticComparativeMethods.pdf>; <https://doi.org/10.1101/2025.06.16.659089>). In this study, "the bladj function from phylocom... estimates node ages based on fossil calibration points while unknown ages are evenly distributed between known nodes" (Lines 411-412), which may introduce inaccurate estimation of branch lengths. The potential effect of this uncertainty in branch length on ancestral state reconstruction should be assessed.

KLH: Thank you for the comment. To address it, we conducted sensitivity analyses by incorporating uncertainty into our ancestral state reconstruction. We generated branch length perturbations using a set of trees with jittered branch length (N = 50, with a 10% perturbation factor) and re-estimated ancestral CT_{max} values across replicates. From these replicates, we calculated mean estimates and standard deviations for the major nodes. Means of ancestral states of CT_{max} were stable across the ensemble with an average change across insect orders of 0.07% and standard deviations were low (details in Table 7) such that our conclusions were

(iii) Pagels lambda as a proxy for the phylogenetic autocorrelation in the data
Full data: Pagel's $\lambda = 0.76$ ($p < 0.001$), Blomberg's $K = 0.33$ ($p < 0.001$); 10,000 randomizations)

Matching families: Pagel's $\lambda = 0.87$ ($p < 0.001$), Blomberg's $K = 0.39$ ($p < 0.001$); 10,000 randomizations)

(iv) the theta parameter estimating the evolutionary optimum in CT_{max}

Full data: 42.20 °C, Matching families: 43.48 °C

3. Discrepancies in family coverage. I believe we can safely assume that the family coverage between the adopted family tree and the barcoded insects is not identical. However, it is not clear how the authors dealt with this. What happened when (if) the new samples contained families that were not included in the 2014 phylogeny, or vice versa? Does this discrepancy change the results and conclusions on CT_{max} evolution?

KLH: We thank the reviewer for this comment on family coverage and agree that more methodological details help to improve the manuscript. In case of families that are included within the backbone tree but not within the samples, these were dropped from the tree before adding the subtrees, following the protocol from Kortmann et al. (2025). Vice versa, for families (n=12 families with 14 samples from a total of 242 families) not included in the backbone tree, these were added manually by placing them next to a sister family included in the backbone. The number of missing families was very low regarding the breadth of sampling of our study (which was the main reason for choosing the Rainford et al. (2014) tree as the most complete backbone tree currently existing). We list here and in the Supplementary Methods the (1) missing families and (2) the closely related families where they were placed within the tree:

Ripterygidae - Tridaetylidae
Ischnorhinidae - Clastopteridae
Cyrtocortidae - Cydnidae
Neridae - Micropezidae
Lyciscidae - Leucospidae
Diparidae - Tetracampidae
Sparasionidae - Scelionidae
Photinidae - Liturgusidae
Lonchodidae - Phasmatidae
Curtonotidae - Drosophilidae
Chirocpteridae - Metalliticidae
Cybocephalidae - Sphindidae

These missing families were minor and summed up to a total of only 14 samples, making up less than 1% of the data. We now show that the removal of manually added families does not change the results on the reconstruction of ancestral states of CT_{max} (average change of 0.04%) here in Table 5 and for the readership in the Extended Data Table 2.

Table 5: Ancestral state CT_{max} values for insect orders and major families for the full data and a subset with manually added families excluded.

Group	Ancestral state CT_{max}	
	Full data	Manually added families excluded
Diptera	40.26145	40.26615
Hymenoptera	42.26314	42.27883
Lepidoptera	41.31444	41.31796
Coleoptera	42.01704	42.02263
Hemiptera	41.81403	41.83760
Orthoptera	43.41890	43.51575
Braconidae	42.53200	42.53543
Chrysomelidae	44.07428	44.07438
Formicidae	44.90566	44.90610
Ichneumonidae	41.71519	41.71846
Muscidae	41.05457	41.03492

We found the same values for full data and for the subset with manually added families excluded: Pagel's $\lambda = 0.76$ ($p < 0.001$), Blomberg's $K = 0.33$ ($p < 0.001$, 10,000 randomizations), theta parameter = 42.20 °C.

4. Monophyly of pre-defined families and barcode sub-trees. Families included in the 2014 phylogeny are presumed monophyletic in the present study. However, taxonomic revision, including those at the family-level, is expected to be a persistent endeavor. It is inappropriate to intuitively assume that the family definition remains unchanged over the past decade. In cases where a pre-defined family is non-monophyletic, the corresponding node should not serve as an anchor point to which the barcode tree is to be attached.

KLH: Thank you for this critical comment. We have searched for non-monophyletic families in the most recent phylogenetic analyses for each insect order, which are the following:

Coleoptera in Cai et al. (2022):
Linnichidae, Elateridae, Geotrupidae, Scarabaeidae, Rhinorhipidae, Melandryidae

Hymenoptera in Blaimer et al. (2023):
Eupelmidae, Pteromalidae, Aphelinidae, Diapriidae, Figitidae, Cynipidae
Lepidoptera in Kawahara et al. (2023): strong support for the monophyly of all families
Orthoptera in Song et al. (2020): All families monophyletic
Hemiptera in Song et al. (2024): All families monophyletic
Diptera in Lambkin et al. (2013): All families monophyletic

The proportion of polyphyletic families was relatively small: From the 12 (6 Coleoptera, 6 Hymenoptera) potentially non-monophyletic families our dataset included $n=5$ families and $n=156$ samples (116 samples from Scarabaeidae). To test the robustness against the inclusion of potentially non-monophyletic families we added additional analyses in which we removed

all individuals of polyphyletic families from the dataset. We tested if results concerning the reconstruction of ancient trait states in CT_{max} or the trends of CT_{max} with elevation were affected by these changes (Fig. 3). We added additional sentences and tables showing the robustness of results against these changes in the phylogenetic tree and dataset in the Supplementary Methods, in the Extended Data Table 2 and Extended Data Fig. 3. We found that the removal of non-monophyletic families has no impact on the results with an average change of CT_{max} estimates of 0.04% (details in Table 6).

Critical families and number of individuals removed:

Diapriidae, 9 individuals
Elateridae, 22 individuals
Eupelmidae, 2 individuals
Pteromalidae, 7 individuals
Scarabaeidae, 116 individuals

Fig. 3: Elevational trends of (a) full data CT_{max} and (b) without non-monophyletic families. (c) Full data CT_{min} and (d) without non-monophyletic families. Black = Afrotropics, grey = Neotropics.

Table 6: Ancestral state CT_{max} values for insect orders and major families for the full data and a subset with non-monophyletic families excluded.

Group	Ancestral state CT_{max}	
	Full data	Non-monophyletic families excluded
Diptera	40.26145	40.28112
Hymenoptera	42.26314	42.27318
Lepidoptera	41.31444	41.32917
Coleoptera	42.01704	42.11637
Hemiptera	41.81403	41.82070
Orthoptera	43.41890	43.42389
Braconidae	42.53200	42.53115
Chrysomelidae	44.07428	44.08519
Formicidae	44.90566	44.90555
Ichneumonidae	41.71519	41.71438
Muscidae	41.05457	41.05457